# The Study of the Relationship among GCI, GII, Disruptive Technology, and Social Innovations in MNCs: How Do We Evaluate Financial Innovations Made by Firms? A Preliminary Inquiry

Aurel Burciu [1,*], Rozalia Kicsi [1], Simona Buta [1], Mihaela State [1], Iulia Burlac [1,2], Denisa Alexandra Chifan [1,3] and Beatrice Ipsalat [1]

1 Department of Management, Business Administration and Tourism, Faculty of Economics and Business Administration, "Ștefan cel Mare" University of Suceava, 720229 Suceava, Romania; rozaliak@usm.ro (R.K.); simonab@usm.ro (S.B.); mihaela.state@usm.ro (M.S.); iulia.burlac@usm.ro (I.B.); chifan.denisa@yahoo.com (D.A.C.); ipsalationelabeatrice@yahoo.com (B.I.)

2 Department of Organisation of Companies and Commercialisation, Faculty of Economics and Business Administration, University of Santiago de Compostela, 15705 Santiago de Compostela, Spain

3 Department of Management and Economics, Faculty of Social Sciences, University of Beira Interior, 6201-001 Covilha, Portugal

* Correspondence: aurel.burciu@usm.ro

**Abstract:** This study aims to assess and identify the role of disruptive/digital technologies in financial innovation strategies as part of social innovations at both the firm and country level. The analysis proposed by the present study brings useful theoretical/pragmatic insights on the application of financial technologies in the context of the "fintech" revolution, as a disruptive innovation. There are few studies of this type that "cross-examine" technical/social innovative capacity at the firm level vs. the same innovative capacity at the level of the world's major countries. Our proposed study brings some novel elements to the literature on this topic. First, the study synthesizes the factors/variables explaining technical/social innovative capacity as ranked by the GCI (Global Competitiveness Index) and GII (Global Innovation Index) at the country level and then correlates informal/empirical variables with the factors explaining innovative capacity for the 50 companies in the BCG (Boston Consulting Group) ranking. Second, the study identifies three "driving forces" (digital technologies, managers, and the market) as the main variables determining financial innovativeness (fintech revolution) at the firm level. Third, based on the "over-cross assessment" (non- statistical) of the information/data provided by the BCG study vs. the GII and GCI studies, the study suggests some ways to delineate and quantify financial innovation as part of social innovation (e.g., it is argued that up to 80% of the social innovation achieved annually by a firm relates to the financial relationships engaged by the firm with various categories of stakeholders). Finally, the study is also important from a pragmatic point of view as it suggests/proposes a number of principles that can be considered by managers for building a KM (knowledge management) and continuous financial innovation strategy. From a theoretical perspective, the study provides a starting point for further research aimed at explaining firm-level financial innovation (fintech as a disruptor) through the massive use of disruptive technologies.

**Keywords:** disruptive technology; financial innovation; social innovation; fintech revolution; MNCs

---

## 1. Introduction

From the 1990s to the present, the entire business environment in the global economy has entered in a phase of instability and/or successive and increasing change, due to political, technological, social, ecological, cultural, and other factors. Since 1970, Drucker [1] partially anticipated the new realities that were to emerge in the world economy, in that

---

major transformations in technology, industrial structures, education, and public policy would generate a chaotic environment for all firms that operate transnationally. This idea has been confirmed by its existence since 1990, and the Great Global Recession of 2008–2010 came as further confirmation [2] of the unprecedented instability of the business environment. More recently, other major events at the global level (the trade war between the US and China, the social crisis caused by COVID-19, the war triggered by Russia in Ukraine, etc.) show us more and more clearly that we live in a risk society [3,4]. Major turbulences of the type mentioned above over the last three decades have generated chaotic changes in the business environment and especially in the financial markets [2]; only in the case of the social crisis generated by COVID-19 has there been relative stability in the financial markets. The stability of stock markets and financial markets at the international level, visible from 2020 until today, can be explained by the fact that systemic trust has remained relatively stable in the main countries of the world as well as at the global level, between firms/institutions and different international organizations [5]. In 2021, the Global Risks Report ranked epidemics among the top global risks with potentially significant impact, whereas after 2010, the main risk was considered financial failure [6]. According to Chen et al. (2019), there is no standard definition for what "fintech" means and what specific technologies have led to its manifestation as a "revolution" and/or disruptive innovation. The present study aims to provide some clarification on the use of digital/disruptive technologies as a tool to support firms in the realization of fintech innovations.

In the context described, given the unstable nature of the capitalist system, innovation can become a creative response to trends and transformations taking place in the economic and social environment [7–10]. Added to this is the technological opportunity created by progressive research in areas such as advanced manufacturing, robotics, and digital technologies and their implementation in different socio-economic areas. In contrast, the turbulence that spread throughout the economy during the economic crisis of 2008 [2] affected the innovative behavior of firms [11,12], especially as a result of delaying access to financial resources, all the more so as the nature of innovation projects makes their financing different from the financing of ordinary assets [13].

This study proposes an analysis of the relationship between financial innovative capacity at the country level (according to GII and GCI) and financial innovative capacity at the firm/company level (according to BCG and Forbes), taking into account that any entity can make extensive use of various disruptive technologies. In relation to the above-mentioned purpose, it is easy to deduce that there are several subsidiary objectives that are considered by the authors:

■  A first objective is to conduct an in-depth analysis at the level of major countries (based on the rankings given by GCI and GII) to understand which are the main factors/pillars that highlight the use of disruptive financial technologies (associated with fintech) to explain technical, social, and financial innovative capacity.

■  Another objective is to conduct a descriptive analysis at the level of MNCs considered to be innovative and high-performing (BCG and Forbes studies) to understand which are the main factors explaining the technical, social, financial, and other innovative capacities of these corporations. Associated with this objective, the study considers some generalizations from MNCs to the SME sector in the context of the fintech revolution over the past decade [14] and to outline/describe potential directions in which various disruptive financial technologies will be used in the future.

Finally, another objective related to the purpose of the present research is to analyze on the basis of "crossover assessment", from countries to firms and vice versa, what has been and is the role of disruptive financial technologies in the context brought by fintech. With regard to this objective of the study, it should be stressed that it does not aim to provide a single/unitary answer to the question in the title "How do you evaluate financial innovation made by firms", given the existence of millions of firms in the global economy. Also, in connection with this third objective of the study, it should be mentioned that the "crossover assessment" envisaged by the authors is to be made only on the basis of an

informal and economic analysis (not on the basis of descriptive statistics). This is because, as Krugman argues [15], countries are "closed socio-economic systems" and firms are "open socio-economic systems"; any crossover assessment of macrosocial vs. organizational perspectives must be approached with great caution, whatever the subject of analysis of the research.

In a turbulent and/or chaotic business environment, the use of digital technologies as part of disruptive technologies can assist firms to improve their technical and social innovative capacity, and thus better respond to the challenges of going through a downswing in the cyclical evolution of business. More specifically, this study aims to identify the clearest and most substantiated principles that would support firms to implement financial innovation (as a major part of social innovation) through the extensive use of disruptive technologies.

A more in-depth analysis of what we call social innovations, and in particular disruptive social innovations, is needed, taking into account existing conceptualizations of disruptive innovation in general. By definition, continuous innovative activity at the firm level assumes the acquisition and processing of new knowledge by skilled employees who are motivated to learn persistently. A paper published in 1995 by Nonaka and Takeuchi argues how tacit and explicit knowledge held by employees in firms is transformed into innovations and patents, respectively, but it is particularly concerned with technical innovations [16]. By explicit knowledge we mean knowledge that exists in books and manuals and can be easily transferred to others. By tacit knowledge we mean knowledge of an intuitive nature, based on experience and which is more difficult to transfer to others.

The structure of our proposed research includes a literature review section, followed by a research design section. In this third section of the study, we formulate some hypotheses of the study and a logic flowchart. In Section 4 of the study, we carry out an in-depth analysis of the factors/variables explaining the competitive position and/or innovative capacity according to the GCI and GII rankings for the main countries of the world. Subsequently, in Section 5 of the study, we present the company approached as a "system" and the financial relationships it engages in with various stakeholders, relationships that can be managed efficiently based on digital technologies. In the final part of the study (Section 6), we present some "driving forces" for financial innovation using disruptive technologies at the firm level and analyze the main variables/factors that explain financial/social innovative capacity for the 50 companies included in the BCG ranking. Also, in the sixth part of the research, we recurred to our own analysis in which we "cross-reference" (mix) the main data provided by the international literature, the GCI study, and the GII study, including the BCG study, in an attempt to suggest our own way of assessing financial innovations as part of the social innovations that are carried out by firms. In this stage, we have outlined a number of "$n$" principles that can be considered for strategic thinking on social innovation at the firm level in different countries of the world. The same principles for financial/social innovation are also of interest from a theoretical perspective as they provide a "common denominator" for future studies that aim to argue the relationship between disruptive technologies and innovative capacity at the firm/country level.

## 2. Literature Review

One of the main ideas of Schumpeter's thought explains economic cyclicality as the result of innovation, which in turn is shaped by economic dynamics [17]. In Schumpeter's view [18], entrepreneurs successively bring technical, organizational, or other novelties to the market and society, which means technical and social innovations. The competition between entrepreneurs and continuous innovation generates, under certain conditions, the emergence of a new industry, which means "creative destruction" [18,19].

In the 1980s, Drucker and other authors argued quite well that social innovations are at least as important as technical innovations [20,21]. Thus, from the 1980s to the present day, we discuss technical innovations (mainly concerning products and technologies) and social innovations (mainly concerning market relations and organization). In 1997,

Christensen [22] proposed the concept of "disruptive innovation" as equivalent to "creative destruction"; subsequently, dozens of volumes and articles have been written on disruptive innovations/technologies and their role for the economic progress of countries/firms [22–28]. There is no single or unified approach to defining the concepts of "technical innovation" and "social innovation" and the elements that give content to each concept. In the same vein, we would point out that the international literature does not provide a clear and uniform method for quantifying "technical innovation" vs. "social innovation", especially when it comes to evaluating/measuring the results of such innovation on social progress.

In organizational studies, Grimm et al. (2013) [29] argue social innovation can refer to social capital, organizational learning and employee training, knowledge sharing, and other aspects that strengthen the capacity of firms/organizations to become more resilient to the changes that a completely chaotic business environment generates [2]. Social innovation, argue Pot and Vass [30], has become one of the major challenges facing Europe today; this type of innovation refers to vision and management structures, organizational flexibility, the development of skills and competences, networking with other organizations, and any other aspect whereby technological innovations are embedded in social structures and contribute to social progress. Most studies on social and/or technical innovation converge towards the conclusion that the widest possible use of digital technologies directly supports companies in achieving core competences [30–32]. At the same time, the use of digital technologies can be located at three levels of analysis, i.e., the macro-level, meso-level, and micro-level [29]; these technologies directly enhance the achievement of technical and social innovations alike. However, it is extremely difficult to quantify precisely the effect/benefits brought about by technical and social innovations at any of the three levels of localization. Even with reference to technical innovations, for which we have the number of patents a firm has annually (by aggregation at the national/regional level), which we also refer to through the sample of 50 innovative companies in the BCG study (Table S2), it is difficult to estimate the commercial effects and impact of a higher number of patents on a company's performance. As shown in Table S2, companies such as IBM obtain up to 9000 patents annually, which gives a fairly clear picture of the R&D investment and creative capacity of the organization (but only a quarter to a third of the annual patent portfolio has commercial application). Even more so, when looking at social innovations at the micro-, meso-, or macro-level, it is almost impossible to quantify precisely the benefits that such innovations bring (both for the world's leading countries and for MNCs entering rankings such as Forbes 2000.

According to the Pareto principle, regarding the random distribution of the results that a process or an economic variable generates, one can accept as a starting point the 80–20% rule in trying to estimate the importance of social innovations vs. technical innovations as an influence on social/economic progress. The same Pareto principle can be used, also as a starting point, when aiming to make a clear distinction between financial innovation and other types of social innovation (which we refer to in more detail from Section 4 of the study to the conclusions part). According to the arguments made by Pot and Vass [30], social innovations account for about 75% of the cumulative effect of innovations on social progress. In Figure 1, we present this delimitation between social innovations and technical innovations as a preliminary argument to the question in the title of the study: "how do we evaluate financial innovations made by firms?"

In relation to the Pareto principle we have invoked, it should be underlined that it is not true in all cases in the global economy and that it is not necessary to confirm it in the activity of thousands or dozens of firms (depending on the field in which a firm is located, there will be situations of the distribution of results such as 60–40%, 50–50%, etc.). Even if the pragmatic usefulness of the Pareto principle is limited, its acceptance can support the extension of fintech theories and the construction of effective KM and continuous innovation strategies at the firm level.

| Technological innovation | Social innovation |
|---|---|
| -Technological Knowledge<br>-R&D and ICT investments<br>-Research and Development<br>-Knowledge creation | -Management Knowledge<br>-Education and experience<br>-Organization, management labour<br>-Acquisition and application of new knowledge |
| Explains 25% of innovation success | Explains 75% of innovation success |

**Figure 1.** Impact of technological and social innovation on social progress. Source: [30].

In direct connection with the distinction between social innovations and technical innovations, it is appropriate to further recall/define concepts such as "disruptive innovation", "creativity", "fintech", "open innovation", "financial innovation", etc. By disruptive innovation we refer to a technical and/or social innovation carried out by a company/organization that brings an element of novelty to the product/service, organization, or market, which then generates major changes in the relationship with consumers and, as a consequence, challenges established companies in the field [22]. Whatever the type of innovation carried out by a company, other organizations, or an individual, at the core of the process of bringing an element of novelty is tacit knowledge and individual and/or group creativity. In a general sense, creativity is the ability to think in new ways in relation to already known problems and to formulate new questions through 'thinking outside the box' processes. The concept of 'fintech' has emerged relatively recently (last decade) in the strategies/practices applied by some firms using digital technologies and bringing major innovations to segments of the financial markets in the global economy. Some authors [14] estimate that, as of 2017, we must discuss a real revolution in relation to the implications of the fintech concept and/or major transformations in financial services. This is because fintech has become a disruptive innovation with major effects on the functioning of traditional financial markets; the fintech concept is challenging banks, insurance companies, investment funds, management consulting firms, etc. The disruptive effects of fintech, according to Gomber et al. [14], take the form of new business models, new markets, new ITC networks, cross-border innovations, new forms of lending, cross-border payments, open banking operations, etc. In summary, it can be said that fintech greatly extends the application of financial technology to individuals, start-ups, thousands of SMEs, etc. [27,31]; fintech is therefore a kind of disruptive social innovation involving major benefits/transformations that may prove to be more socially important than technical innovations on products [29]. In direct connection with disruptive innovation, the concept of open innovation has developed more recently [32,33], which essentially refers to the orientation of a firm towards various stakeholders to build alliances, partnerships, and other business networks through the use of digital technologies, leading to a cumulative effect on the sources for continuous innovation. In regards to the concept of "financial innovation", it refers to any innovation that an organization or country manages to bring about with regard to processes, operations, or activities involving financial flows either of the traditional type (through banks, insurance companies, investment funds, etc.) or of the digital type, among which fintech operations and/or markets already have a well-defined position. The expansion of fintech firms has relied, in particular, on "unbanked costumers", various start-up firms in high-tech industries, and millions of investors from various countries; a new business model applied by fintech firms through the digitization of financial services has given rise to new market segments. From the perspective of studies on fintech, as assessed by Aysan

and Nanaeva [34], fintech has become a financial disruptor; most studies on this topic, a total of 363 papers, have only been published since 2017. It will be very difficult for large corporations, argue Gomber et al. [14], to build strategies in KM and with respect to the acquisition of new knowledge and Human Resources (HR) to be competitive in fintech markets where small start-ups that rely exclusively on disruptive financial technologies are already operating.

In some situations, Christensen argues [22], large companies remain somehow captive "by their customers", they react slowly and the new markets that emerge are quickly filled by successful start-ups that build their strategy on the use of digital technologies. According to a study by Deloitte [35], traditional financial institutions are increasing their investments in fintech operations/activities year after year, particularly in the acquisition of new disruptive technologies, in order to compete with fintech start-ups based on cloud computing, mobile telecommunications, and similar technologies. The same Deloitte study anticipates that there are several important trends in the fintech markets (trend 1: traditional financial firms are increasingly involved in fintech operations; trend 2: blockchain technology eliminates the need for intermediaries in various asset transactions; trend 3: in various countries, governments and other regulators are increasingly interested in regulating fintech markets; and trend 4: financial company executives are increasingly interested in the challenges that disruptive technologies bring). According to some assessments made by fintech industry experts for the year 2023 [36], especially following the bankruptcy of the American company FTX, new regulations will be adopted in the fintech markets, but fintech technologies will continue to remain popular in the future (such predictions are also found in some units/services of large corporations such as IBM, according to financial services digital transformation). Regional and/or country differences in fintech adoption exist and will remain, but all assessments indicate that the fintech industry will continue to expand rapidly in the coming years [37].

The disruptive or non-disruptive nature of a technical, social, or financial innovation cannot be anticipated by innovators, firms, or organizations; it will be determined by the confrontation between the outcome of the innovation and the market and/or society. When discussing the practice of innovation, says Drucker, one can only speak of intentional innovation as the result of a systematic analysis/effort made by individuals and organizations [21]. In some cases, companies may imitate other competitors with respect to technical/social innovations that are made and already have acquired market effects [20]. the orientation of Chinese firms towards "market-driven R&D" and/or "knowledge-driven R&D" has allowed some MNCs in China to develop "open innovation" networks and become formidable competitors in some international markets. Chinese companies such as Alibaba or Yuwell, say Yip & McKern, have realized their own social innovations and started to develop separate divisions for fintech operations/markets [20]. The orientation of firms towards both technical and social "open innovation" networks, the formation of various alliances/partnerships, and the extensive use of disruptive financial technologies directly support KM application strategies for continuous innovation [38,39].

Some studies highlight the role of digital technologies in enhancing sustainable development in various countries/regions of the world, in balancing gender ratios, in managing various organizations, in holding social positions, etc. [40–43]. Other studies explore, as appropriate, the role of digital/disruptive technologies for GDP growth at the country level, for process optimization at the company level, as well as on the changes these technologies bring to society; times of crisis in society/economy seem to be better managed through ICT-enabled advances [8,13,44–48]. In short, it can be concluded that episodes of crisis transform the environment and the innovative behavior of firms. Thus, while the pre-crisis model of creative accumulation better explains the results, in the post-crisis period, the model of creative destruction seems to dominate. Moreover, the manufacturing sector, the main generator of technological innovations, tends to matter less in value creation and employment, while the services sector is more likely to compete through non-technological innovations.

## 3. Research Design

### 3.1. Hypotheses and Stages

In our study, we aimed to identify/argue the relationship between disruptive technology and financial innovations (related to fintech) simultaneously at the level of major countries of the world and performing MNCs in international competition. At the same time, we intend to suggest some principles for the realization of financial innovation by firms; such principles would also be of theoretical interest in that they would become a starting point for other similar studies. Most studies [43,49] on financial innovation through the use of disruptive technologies refer either to firms in high-tech sectors of the manufacturing industry (pharmaceuticals, ITC, etc.) or to firms in more knowledge-intensive service sectors—KIS (especially in the banking sector, insurance sector, and other fintech start-ups; [36]). There are few studies that propose financial innovation evaluation by technologies from both a country and a firm perspective and that, in addition, propose a synthetic analysis of the factors/principles that explain the innovative capacity of an entity (whether it is a firm in the manufacturing industry or KIS). Another important aspect to achieve the aim of our research is the identification of a unified classification of innovation types/categories. In the sense invoked, according to the Oslo Manual under the auspices of the OECD [50] and the European Commission, there are four types/categories of innovations: product innovations, process innovations, marketing innovations, and organizational innovations (ref. [50]). As we will show later (point 6.2 in Section 6 of the study), financial innovation at the level of any firm refers, in particular, to marketing innovations and organizational innovations (even if it is not possible to clearly delimit/dissociate between the four types of innovations mentioned, the way in which protection can be obtained for an element of novelty in a firm, and the industry sector in which the firm is located). In direct connection with the intended purpose of the study, it aims to conduct an assessment/analysis from the SME sector to established MNCs in the context of the revolution brought by fintech firms and/or markets (ref. [14]) in order to identify strategies/practices that large corporations (non-financial and financial), according to UNCTAD classification [51], are resorting to in order to adapt to the new realities generated by financial technologies. Also, in the sense mentioned above, we aim to understand the conceptualizations that have recently emerged in the literature on fintech and the directions that can be envisaged regarding the quantification of financial innovations based on disruptive technologies.

In order to achieve the proposed objectives, we have proceeded through several steps/stages specific to such research:

✓ In stage 1 (abbreviated S1), we selected and reviewed the international literature on the concept of fintech and on financial innovation, the use of digital technologies in society, etc., both from a macro-social/societal perspective and from the perspective of firms.

✓ In stage 2 (abbreviated S2), at the basis of our study, we state the following four research hypotheses:

**H1:** There is no direct correlation (causality, association, etc.) between annual technical innovation, in terms of the number of patents, and annual social innovation (trademarks, industrial designs, etc.; social innovations related to strategy and organizational structure and which cannot be protected with registration) at the level of MNCs or SMEs. The sector in which a firm is located (high-tech, medium-tech, low-tech, etc.) provides opportunities/conditions for firms to innovate towards technical innovations, social innovations, or both. However, it remains essential for the top-management vision of any organization, MNCs, or SMEs, regarding the achievement of technical vs. social innovations through the acquisition and processing of new knowledge.

**H2:** In the overall social innovation carried out annually by a firm, MNCs, or SMEs, financial innovation may account for up to 80% of the novelty elements brought by the organization in relation to its organizational structures, market, or society. This hypothesis is taken into account in the study, since most of the current relationships (daily, monthly, etc.) that a firm

engages in with various stakeholders (shareholders, suppliers, customers, banks, etc.) also involve financial relationships. It is neither possible nor necessary to quantify precisely the share of financial innovations as part of annual social innovations, as there are millions of firms in the global economy. The widespread application of digital technologies in most countries of the world and the almost exponential growth of interest in the concept of fintech is a relatively current reality, respectively, from 2018 to the present, with particular reference to financial industries. It is useful and beneficial to clearly differentiate between financial innovation and other social innovations based on the Pareto principle for the random distribution of outcomes of economic variables.

**H3:** Countries that are internationally competitive/innovative have a number of MNCs and/or SMEs that are each at least average in innovation and annual performance (both technical and social). This is even if only a small number of MNCs from various countries end up being included in international rankings pre-like BCG, Forbes, etc. The "over-cross assessment" of the influence relationship between firms and countries with respect to annual technical/social innovation cannot be performed on the basis of descriptive statistics. This assessment can and should only be made on an empirical basis, taking into account studies on fintech and financial innovation [14,29,31,37,52], together with a logical and cross-sectional analysis of the information provided by some firm rankings (BCG, Forbes, etc.) vs. some country rankings (GCI, GII, etc.).

**H4:** The widest possible use of disruptive technologies (especially digital ones) greatly enhances the innovative capacity of any entity (country, company, and other organizations), both for technical and financial/social innovations. All the international rankings on which the present study is based (GCI, GII, BCG, etc.) include in their calculation methodology at least two or three sub-pillars that take into account the use of ITC by firms or individuals at various levels of a reference/analysis. This is particularly because digital technologies have become essential today for education, access to knowledge, increasing labor productivity, continuous innovation, open innovation networking, etc.

✓  In stage 3 (abbreviated S3), we selected the GCI (Global Competitiveness Index) ranking [53,54] for the period 2009–2019 in an attempt to obtain a first picture of the competitiveness/innovative capacity of the world's major countries. This first picture based on the GCI gives us, at the same time, elements of interest for our study on the innovative capacity existing in the firms of these countries (since some sub-pillars such as social capital, cooperation in labor employer relation, university industry collaboration in R&D, and the extent of staff training give us important information for innovation in firms). In the case of this GCI ranking, we selected only 28 main countries of the world for which we present in Table S1 the preliminary statistical data on which we subsequently applied specific assessments such as a factor analysis, regression analysis, etc. According to the summary in Table S1, it appears that we selected at the same time only seven sub-pillars considered as main and which provide relevant information for the cross analysis of innovative capacity, simultaneously at the level of firms and countries. In the case of the GCI ranking, we selected 2019 vs. 2009, and then in the case of the GII ranking, we selected the ranking for 2020 vs. 2010 since, for the purpose of the proposed research, an evaluation based on descriptive statistics between the two different rankings is not envisaged. The assessment is intended to be performed only empirically and economically between the information provided by the GCI and the GII for the factors that support/explain innovative capacity in firms (part 2 of the proposed study).

✓   In stage 4 (abbreviated S4), we selected the GII (Global Innovation Index) ranking [55,56] for the period 2010–2020 to see which of the world's major countries have an innovative capacity above the average of the entire group of countries analyzed by this annual ranking. Even the choice of this ranking was determined by the fact that some of the sub-indexes (e.g., gross expenditure on R&D and global R&D companies; ICT access, ICT use, GERD performed by business enterprise, etc.) also provide valuable

information on the innovative capacity of firms in these countries. The inclusion in our study of GCI and GII at slightly postponed times has been foreseen for informal comparisons that can be formulated already at this stage of the study (year 2010 vs. 2009, respectively, year 2020 vs. 2019), in order to connect these comparisons later with the microeconomic perspective on innovative, technical, and/or other capacities in the main countries/firms of the world.

✓ In stage 5 (abbreviated S5), we chose a study provided by BCG (Boston Consulting Group) [44], which highlights 50 of the most innovative companies in the world (27 firms—USA; 15 firms—Asia; and 8 firms—Europe) and for which, based on the Annual Report of each organization, we highlighted, in Table S2, technical and social innovative capacity together with some information on size, financial performance, etc. The ranking provided by BCG (summarized in Section 6 of the study and excerpted in Table S2) is for the years 2022 and 2023, but has been calculated annually by the Boston consulting firm since 2004. The data on the 50 MNCs in the BCG study were then statistically ordered with a clustering analysis of the firms in the sample, drawing dendograms for the selected firms, etc., trying to identify association relationships between companies, sectors in which they are located, net income, and countries of affiliation. In addition to the information provided by the BCG study, we analyzed in depth the annual ranking published by Forbes, namely Forbes 2000 [45], to see, selectively, the association between some of the less-known companies, their inclusion in the fintech category, and their countries of origin.

The evaluations in the S5 stage started with the S3 stage of the study and ran continuously, simultaneously, and in parallel with other stages of the study such as S4, S6, S7, S8, and S9, as we present graphically, in the next section of the study, under a flowchart underlying our research.

✓ In stage 6 (abbreviated S6), we conducted, in parallel with S3, an analysis based on descriptive statistics on all the information provided by the seven sub-pillars of the GCI ranking; this information was then empirically and not statistically correlated with similar information provided by the GII and that based on descriptive statistics in the BCG ranking.

✓ In stage 7 (abbreviated S7), similarly to the previous step, but in parallel with S4, we carried out an analysis also based on descriptive statistics on all the information provided by the GII ranking; this information on innovative capacity at the level of the main countries was then correlated empirically and not statistically with similar information provided by the GCI and that based on descriptive statistics in the BCG ranking.

✓ In stage 8 (abbreviated S8), we conducted an in-depth analysis based on descriptive statistics for only the 50 companies considered by BCG to be the most innovative globally in 2022. Including for this ranking, we empirically and informally analyzed the innovation situation at the time of 2023 vs. 2022, but the static tests of correlation, clustering, etc., and the data in Table S2 are only for the 2022 ranking. It is not possible to make a statistical analysis of the relationship between the innovative capacity at the firm level (the 50 firms in the BCG ranking) and the innovative capacity at the country level; this relationship can only be described and interpreted empirically and informally.

✓ In stage 9 (abbreviated S9), we include, as a separate step, a sequential "crossover assessment" between factors/variables, showing financial innovation as part of social innovation on the relationship between countries and firms and vice versa. Including in this S9 stage, no statistical tests were considered and applied to analyze the relationship between the information provided by the BCG survey and the information provided by the two country rankings.

### 3.2. Logical Framework of the Study

The international literature on fintech firms/organizations [14,31,37] shows us that particularly in the case of SMEs, there has been a strong interest in applying digital technologies to develop/expand mainstream and/or disruptive financial innovations from 2016 to the present. This topic related to fintechevolution was continuously considered by the authors in the development/study of all nine stages of the study. The flow chart of the study is presented in Figure 2.

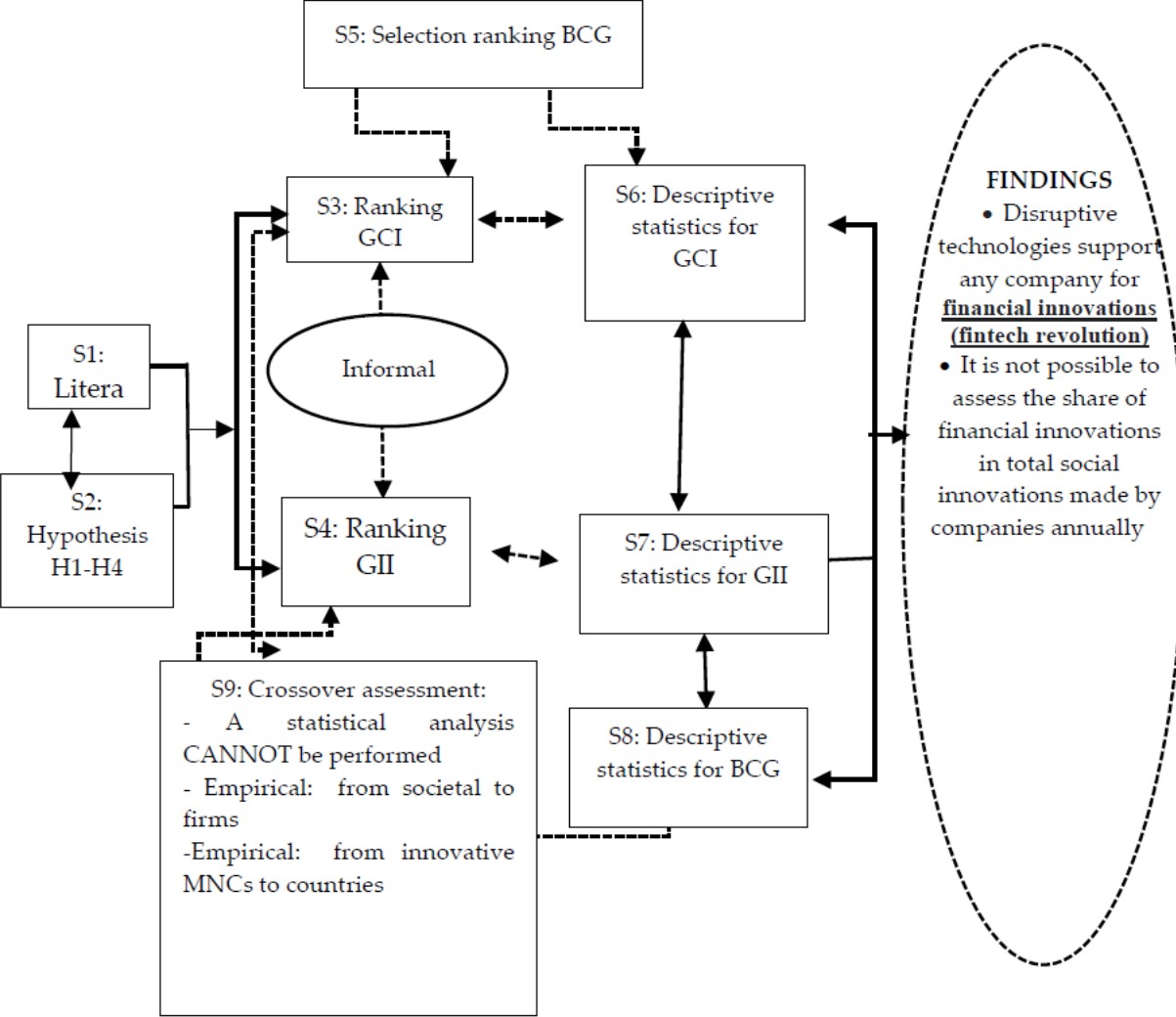

**Figure 2.** The logic flowchart of the study. Source: elaborated by the authors.

In direct relation to this section of the study (Section 3), we formulate two mentions related to the research methodology:

a. First, the analysis based on "descriptive statistics" is applied only for the processing/interpretation of data "inside" each in-country ranking, i.e., GCI, GII, and distinctly BCG, not for the "crossover assessment" on the identification of factors explaining the financial/social innovation capacity simultaneously at the firm and country level. The hypothesis of applying a crossover assessment based on descriptive statistics from the BCG study to the two country rankings was considered by the authors, but it leads to non-rational conclusions that contradict the realities of the global economy (it would be inferred that only US firms, which categorically dominate such rankings in the proportion of about 2/3 of the total ranked firms, are the most fi-

nancially/socially innovative and that only the culture of these firms supports the application of KM strategies).

b.  We used a double-cross analysis of the information provided by the GCI and GII rankings reflecting the technical/social innovation capacity of the world's major countries. Thus, in the case of the GCI ranking, we first selected 7 sub-pillars (out of 12 pillars), based on two criteria, considered to be more important for the purpose of the study, together with 28 countries, and then we evaluated this information only economically vs. GII vs. BCG (not statistically). Secondly, in the case of the GII ranking, we performed a descriptive statistics analysis based on the "inside" ranking data and following a clustering analysis, we retained which sub-pillars are more important (along with average/above-average countries) that could explain the financial/social innovation capacity at the firm level. It is neither appropriate nor necessary to perform a crossover assessment analysis based on descriptive statistics between the GII and GCI data, given the intended purpose of the study. It is neither appropriate nor necessary to perform a similar "crossover assessment" based on descriptive statistics between the information provided by BCG for 50 innovative companies and the information provided by GCI and GII (the study argues that a firm is an open socio-economic system and any country is a closed socio-economic system, so they are different realities and should be evaluated with great caution whatever the topic of the research).

## 4. Analysis from Societal Perspective

### 4.1. Implications of GCI for Financial Innovation

By "implications of the GCI for financial innovation" we mean that this ranking provides some useful, although partial, information for understanding and subsequently assessing (Section 5 of the study, Main Findings) innovative capacity at the firm level. Our assessment focuses on MNCs, as the realities of the last three decades in the global economy lead to the conclusion that these categorized organizations in particular have become the main vectors for technical and social innovation. As argued in [57], medium and large firms have become essential for R&D and innovation, job creation, exports, revenues, productivity, and other critical indicators for competition in different markets. At the same time, our assessment, reasoning, and conclusions may also include, where appropriate, firms in the SME category (in high-tech sectors such as IT, telecommunications, etc., firms with a significant number of employees can quickly become highly innovative).

Based on this ranking (GCI), we conducted a comparative analysis at the time of 2009 and, respectively, 2019, trying to identify what are the main correlations and significant associations between the different sub-pillars of the ranking, as well as the extent to which these sub-pillars support the understanding of technical and social innovative activity at the firm level (from a microeconomic analysis perspective). As is well known, the GCI ranking is based on 12 main pillars (institution, infrastructure, ICT adoption, macro-economic stability, product market, business dynamism, innovation capability, etc.); countries are grouped by main ranking and by sub-pillars, based on a "score" expressed on a scale of 0–100 (the relative position of countries). In our analysis, we selected the number of seven variables that should show us, cumulatively, the innovative capacity at the country level (abbreviations used: social capital—SC; health and primary education—HPE; health–life expectancy—HLE; higher education and training—HET; mean years of schooling—MYS; extent of staff training—EST; cooperation in labor–employer relations—CLER; state of cluster development—SCD; university–industry collaboration in R&D—UIC-R&D; and multistakeholder collaboration—MC). In relation to the seven variables selected by the authors for the statistical evaluation of possible correlations in 2019 vs. 2009 (data resulting in tables starting from Tables 1–9), we mention:

**Table 1.** Correlation Matrix [a] for the year 2009.

| Variables | | SC | HPE | HLE | EST | CLER | SCD | UIC-R&D |
|---|---|---|---|---|---|---|---|---|
| | SC | 1.000 | **0.500** | **0.496** | **0.402** | 0.305 | 0.035 | 0.289 |
| | HPE | 0.500 | 1.000 | **0.748** | 0.297 | **0.425** | 0.123 | **0.429** |
| | HET | 0.496 | 0.748 | 1.000 | **0.694** | **0.555** | 0.179 | **0.791** |
| Correlation | EST | 0.402 | 0.297 | 0.694 | 1.000 | **0.705** | 0.087 | **0.800** |
| | CLER | 0.305 | 0.425 | 0.555 | 0.705 | 1.000 | **−0.296** | **0.689** |
| | SCD | 0.035 | 0.123 | 0.179 | 0.087 | −0.296 | 1.000 | 0.101 |
| | UIC-R&D | 0.289 | 0.429 | 0.791 | 0.800 | 0.689 | 0.101 | 1.000 |
| | SC | | 0.003 | 0.004 | 0.017 | *0.058* | *0.431* | *0.068* |
| | HPE | 0.003 | | 0.000 | *0.063* | 0.012 | *0.267* | 0.011 |
| | HET | 0.004 | 0.000 | | 0.000 | 0.001 | *0.181* | 0.000 |
| Mr (1-tailed) | EST | 0.017 | 0.063 | 0.000 | | 0.000 | *0.330* | 0.000 |
| | CLER | 0.058 | 0.012 | 0.001 | 0.000 | | 0.063 | 0.000 |
| | SCD | 0.431 | 0.267 | 0.181 | 0.330 | 0.063 | | *0.304* |
| | UIC-R&D | 0.068 | 0.011 | 0.000 | 0.000 | 0.000 | 0.304 | |

[a] Determinant = 0.007.

**Table 2.** Correlation Matrix [a] for 2019.

| Variables | | SC | HLE | MYS | EST | CLER | SCD | MC |
|---|---|---|---|---|---|---|---|---|
| | SC | 1.000 | **0.703** | **0.539** | **0.398** | **0.514** | **0.572** | **0.348** |
| | HLE | 0.703 | 1.000 | 0.466 | 0.164 | 0.296 | **0.450** | 0.124 |
| | MYS | 0.539 | 0.466 | 1.000 | **0.563** | **0.439** | **0.510** | **0.526** |
| Correlation | EST | 0.398 | 0.164 | 0.563 | 1.000 | **0.799** | **0.666** | **0.932** |
| | CLER | 0.514 | 0.296 | 0.439 | 0.799 | 1.000 | **0.563** | **0.738** |
| | SCD | 0.572 | 0.450 | 0.510 | 0.666 | 0.563 | 1.000 | **0.735** |
| | MC | 0.348 | 0.124 | 0.526 | 0.932 | 0.738 | 0.735 | 1.000 |
| | SC | | 0.000 | 0.002 | 0.018 | 0.003 | 0.001 | 0.035 |
| | HLE | 0.000 | | 0.006 | *0.203* | *0.063* | 0.008 | *0.265* |
| | MYS | 0.002 | 0.006 | | 0.001 | 0.010 | 0.003 | 0.002 |
| Mr (1-tailed) | EST | 0.018 | 0.203 | 0.001 | | 0.000 | 0.000 | 0.000 |
| | CLER | 0.003 | 0.063 | 0.010 | 0.000 | | 0.001 | 0.000 |
| | SCD | 0.001 | 0.008 | 0.003 | 0.000 | 0.001 | | 0.000 |
| | MC | 0.035 | 0.265 | 0.002 | 0.000 | 0.000 | 0.000 | |

[a] Determinant = 0.002.

**Table 3.** KMO and Bartlett's Test for 2009.

| KMO and Bartlett's Test | | |
|---|---|---|
| Kaiser–Meyer–Olkin Measure of Sampling Adequacy. | | 0.627 |
| | Approx. Chi-Square | 119.937 |
| Bartlett's Test of Sphericity | df | 21 |
| | Mr | 0.000 |

**Table 4.** KMO and Bartlett's Test for 2019.

| KMO and Bartlett's Test | | |
|---|---|---|
| Kaiser–Meyer–Olkin Measure of Sampling Adequacy. | | 0.780 |
| Bartlett's Test of Sphericity | Approx. Chi-Square | 143.035 |
| | df | 21 |
| | Mr | 0.000 |

**Table 5.** Total Variance Explained for 2009.

| Component | Initial Eigenvalues | Extraction Sums of Squared Loadings | | | Rotation Sums of Squared Loadings | | |
|---|---|---|---|---|---|---|---|
| | Total | % of Variance | Cumulative % | Total | % of Variance | Total | |
| 1 | 3.770 | 53.860 | 53.860 | 3.770 | 53.860 | 3.763 | |
| 2 | 1.234 | 17.623 | 71.484 | 1.234 | 17.623 | 1.238 | |
| 3 | 0.938 | 13.395 | 84.879 | | | | |
| 4 | 0.586 | 8.378 | 93.257 | | | | |
| 5 | 0.258 | 3.684 | 96.941 | | | | |
| 6 | 0.147 | 2.095 | 99.037 | | | | |
| 7 | 0.067 | 0.963 | 100.000 | | | | |

Extraction Method: Principal Component Analysis. (When components are correlated, sums of squared loadings cannot be added to obtain a total variance).

**Table 6.** Total Variance Explained pentru 2019.

| Component | Initial Eigenvalues | | | Extraction Sums of Squared Loadings | | | Rotation Sums of Squared Loadings |
|---|---|---|---|---|---|---|---|
| | Total | % of Variance | Cumulative % | Total | % of Variance | Cumulative % | Total |
| 1 | 4.216 | 60.222 | 60.222 | 4.216 | 60.222 | 60.222 | 3.848 |
| 2 | 1.369 | 19.562 | 79.784 | 1.369 | 19.562 | 79.784 | 2.741 |
| 3 | 0.534 | 7.625 | 87.408 | | | | |
| 4 | 0.435 | 6.212 | 93.620 | | | | |
| 5 | 0.255 | 3.642 | 97.262 | | | | |
| 6 | 0.140 | 1.997 | 99.259 | | | | |
| 7 | 0.052 | 0.741 | 100.000 | | | | |

Extraction Method: Principal Component Analysis.

**Table 7.** Matrix of components rotated at the time of 2009.

| Pattern Matrix [a] | | |
|---|---|---|
| Variables | Component | |
| | 1 | 2 |
| SC | 0.599 | |
| HPE | 0.714 | |
| HET | 0.924 | |
| EST | 0.837 | |
| CLER | 0.759 | −0.541 |
| SCD | | 0.881 |
| UIC-R&D | 0.868 | |

Extraction Method: Principal Component Analysis.
Rotation Method: Oblimin with Kaiser Normalization [a].

[a] Rotation converged in 4 iterations.

**Table 8.** Matrix of rotated components at the time of 2019.

| | Pattern Matrix [a] | |
|---|---|---|
| | Component | |
| **Variables** | **1** | **2** |
| SC | 0.721 | 0.544 |
| HLE | 0.535 | 0.766 |
| MYS | 0.736 | |
| EST | 0.871 | |
| CLER | 0.824 | |
| SCD | 0.839 | |
| MC | 0.852 | −0.464 |

Extraction Method: Principal Component Analysis [a].

[a] Two components extracted.

**Table 9.** Rotated Component Matrix at the time of 2019.

| | Rotated Component Matrix [a] | |
|---|---|---|
| | Component | |
| | **1** | **2** |
| SC | | 0.833 |
| HLE | | 0.985 |
| MYS | | 0.485 |
| EST | 1.001 | |
| CLER | 0.816 | |
| SCD | 0.649 | |
| MC | 1.015 | |

Extraction Method: Principal Component Analysis.
Rotation Method: Oblimin with Kaiser Normalization [a].

[a] Rotation converged in 6 iterations.

In the following, we summarize in Tables 1 and 2 the "correlation matrix" for previously reported issues at the time of 2009 and 2019, respectively.

We present below the correlation matrix in SPSS for 2009 (Table 1) and for 2019 (Table 2). In the first half of the tables are the pairwise correlations between the seven variables included in the analysis for the 2 years (sub-pillars). In the second half of the table are the significance coefficients calculated for the correlation coefficients obtained (having different values in 2019 vs. 2009).

The methodology for calculating the GCI in 2019 has changed significantly compared to the one applied in 2009, which is why some sub-pillars in 2019 have different names from their 2009 counterpart, as they have taken in their content other elements/factors of the same nature (pillar HPE and HLE have different names, but both refer to human capital; similar to HET and MES; similar to innovative capacity, i.e., UIC-R&D and MC).

The selection of the seven sub-pillars was made based on the relevance/significance criteria they cumulatively provide on the factors determining technical and social innovative capacity at the firm level; the same relevance of the selection criterion was taken into account by empirically relating the information provided by GCI vs. GII (the latter is presented by us in the next subsection of the study). A second selection condition was given by the need to "cover" most of the 12 main pillars with the seven variables and which refer, at the same time, directly or indirectly, to R&D and innovation activity in firms. A

total of 28 main countries of the world were selected (countries that are in the top positions in the ranking both at the time of 2009 and at the time of 2019; Switzerland, USA, Japan, Singapore, South Korea, Germany, Denmark, France, the Netherlands, UK, etc.; countries that are important in the global economy but have a more prudent/modest position in the ranking; countries such as Argentina, Brazil, China, India, Mexico, South Africa, etc.).

In both tables, we used some unitary underlining, respectively:

In each table, we marked in bold the significant correlations between the seven variables (this means statistically significant correlations; some sub-pillars help us to further understand the factors explaining the innovative capacity of the companies in the BCG ranking).

We marked in bold and italics in each table the sig. coefficients of significance greater than 0.05 between the seven variables, which indicate that there are no statistically significant correlations.

Taking into account the previous mentions, some conclusions of interest for our study can be drawn (which will then be correlated with the information shown by the GII ranking and the situation of the 50 companies in the BCG ranking):

The situation of insignificant correlations between the seven variables has changed significantly during the decade under analysis, i.e., in 2009, there were five correlations of this type, and in 2019, there were only two correlations of this type:

At the time of 2009, the realities of the following variables were insignificant:

- SC variable and CLER, SCD variable and UIC-R&D;
- HPE variable and EST, SCD;
- HET variable and SCD;
- EST variable and SCD;
- SCD variable and UIC-R&D.

At the time of 2009, a smaller number of the following variables were insignificant:

- HLE variable and EST;
- CLER variable and MC.

In addition, it is easy to notice that the association between variables recording insignificant correlations is completely different at the two moments of analysis (this means that different pillars of the GCI component advanced/evolved differently from one country to another and led to changes in position at both the pillar and ranking level in 2019 vs. 2009).

The situation of significant correlations between the seven variables has changed significantly over the decade analyzed, i.e., in 2009, there were three such correlations and in 2019, there were four such correlations:

✓ At the time of 2009, there were significant differences between the following variables:

  - HPE and HET (0.748);
  - EST and CLER (0.791);
  - EST and UIC-R&D (0.800).

✓ At the time of 2019, there were significant realities between the following variables:

  - EST and CLER (0.799);
  - EST and MC (0.932);
  - CLER and MC (0.738);
  - SCD and MC (0.735).

From the data presented in Tables 1 and 2, it is easy to see that the association between variables with significant correlations has changed significantly during the decade under analysis, 2019 vs. 2009. At the time of 2019, there were four statistically significant correlations between the seven variables compared to only three such correlations at the beginning of the period. These statistical correlation changes between the different variables/factors about which the GCI provides information derive from changes in the overall ranking and/or ranking of the pillars in the competitive position component. Such

changes in terms of the ranking provided by the GCI for the world's leading countries will then be analyzed, "cross-referenced", with the main conclusions provided by the GII ranking (including performing an informal analysis, using main influencing factors, of some deductible aspects for the year 2020 vs. 2019 and 2010 vs. 2009, respectively) and the main conclusions provided by the BCG ranking for the 50 MNCs considered to be the most innovative in the world.

Next, based on the information provided by the same GCI ranking, we calculate and present synthetically the KMO and the Bartlett sphericity test (Tables 3 and 4), as well as the Initial Eigenvalues associated with each factor before extraction, after extraction, and after rotation (Tables 5 and 6).

Tables 3 and 4 present two indicators relevant to our study, namely the KMO (Kaiser–Meyer–Olkin) and the Bartlett sphericity test. The KMO varies between 0 and 1. A value close to 0 indicates that the sum of partial correlations is relatively high compared to the sum of correlations and a factor analysis is not indicated while a value close to 1 indicates that a factor analysis should produce distinct and reliable factors. A value above 0.5 is considered acceptable (http://cda.psych.uiuc.edu/psychometrika_highly_cited_articles/kaiser_1974.pdf; accessed on 11 August 2023). In this case, the value is 0.627, so it is an acceptable value.

Bartlett's test of sphericity is highly significant, the sig. value is less than 0.01, which means that the correlation matrix R is not an identical matrix. There are links between variables that could be included in our analysis based on the GCI ranking.

In the following table, applying the same tests based on data provided by the GCI at the time of 2019 leads to slightly higher values than at the time of 2009.

Tables 5 and 6 centralize the Initial Eigenvalues associated with each factor before extraction, after extraction, and after rotation. Before extraction, seven factors were identified corresponding to the seven variables included in the analysis. The initial values associated with each factor also show the weight of the explained variants. For example, at the time of 2009, the first factor explains 53.86% of the total variance and the second factor explains 17.623%. It can be seen that the first two factors together explain 71.48% of the total variance. After extraction, at the same time of 2009, only two factors remained as can be seen in Table 5 (SPSS extracts—only factors that have values above 1). It can be seen that for the extracted factors, the values are identical to those before extraction. In the last part of the table are the Eigenvalues after rotation, before and after rotation; the first factor and the second factors keep their values.

A more synthetic version of the analysis on the seven variables at the time of 2019 vs. 2009 can be obtained by plotting a Scree plot for the year 2009 (Figure 3) and 2019 (Figure 4).

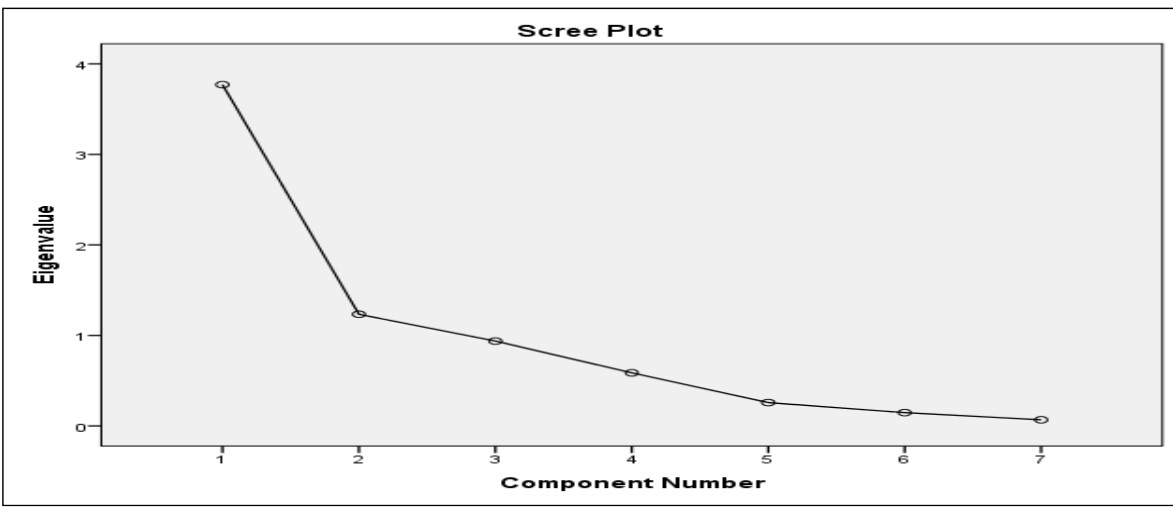

**Figure 3.** Presentation of the 2009 Scree plot.

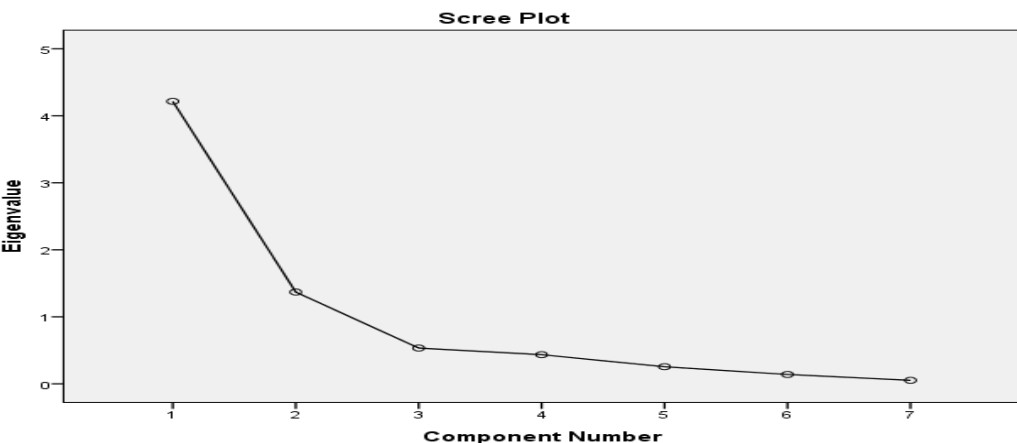

**Figure 4.** Presentation of the 2019 Scree plot.

From Figures 3 and 4, it can be deduced that the main points of inference occur in variables 2 and 3, which means that education, employee training, and R&D investment are the main factors explaining innovative capacity at the country level. This preliminary conclusion has significance for the objective of our research, and we will analyze whether or not various elements related to education, training, and R&D are reflected in the methodology applied by BCG to establish the ranking of the 50 globally innovative companies.

The next step the authors used was to analyze the variables that were assigned in the first factor to see if there is some common theme. It can be seen that between the variable *SCD*, which has the highest loadings distributed in component 2, and the variable *CLER*, there is a certain relationship. Therefore, the variables allocated to the first component at the time of 2009 seem to fall under the same theme, namely education. All of them are related to the same aspect as follows: *HET, UIC-R&D, EST, CLER, HPE*, and *SC*.

Table 7 shows the matrix of rotated components, for the year 2009, then in Table 8, the same data for the year 2019.

Factor loadings less than 0.5 were removed from the Table 7. We can see that there are two components and most of the variables are distributed in the first component, except for two variables: *CLER* and *SCD*, which have been distributed in the second component. The variables are presented in the table in the order in which they were entered. A descending sort by load size places the variable *HET* first, followed by *UIC-RD*.

Factor loadings less than 0.4 were removed from the Table 8. We can see that two components and four variables (EST, CLER, SCD, and MC) are distributed in the first and three variables (SC, HLE, and MYS/MC) are distributed in the second component.

The Multistakeholder_collaboration2019 variable is ranked first by the upload size. Therefore, it can be inferred that, although there have been some changes on the variables/factors explaining the innovative capacity of countries in 2019 vs. 2009, education, staff training, and collaboration with various categories of stakeholders remain essential for innovative capacity at the country level. Finally, the last stage of the authors' analysis of the importance of the variables/factors on which the GCI ranking is based is presented at the time of 2019, as shown in Table 9.

It can be seen that there have been some changes in 2019 in the distribution of variables by components compared to 2009; however, the conclusion remains that social capital, education, staff training, R&D activity, and collaboration with various categories of stakeholders remain the main factors conditioning the innovative capacity of countries. As we will see later (evaluation based on the top 50 BCG rankings [58]), employee education, investment in R&D, firms' orientation towards open networks for innovation, staff training, organizational culture and/or social capital, collaboration between management and employees/trade unions, etc., are the main factors determining innovative capacity at the firm level. It is, however, extremely difficult to argue which would be the "common factors" that simultaneously explain innovative capacity at the firm and country level.

Next, we conduct an analysis based on the Global Innovation Index (GII) to further assess innovative capacity at the country level and the theoretical/hypothetical relationship between such rankings and innovative capacity at the firm level. Later (Section 5 of the study, Main Findings), when we perform an in-depth analysis based on the BCG survey of the 50 most innovative companies internationally, we will try to "disentangle/separate" social/financial innovation from overall firm-level innovative capacity. Also, in the above sense, based on the information provided by the GCI and GII, we will then try to "cross-check" the results we arrive at on financial innovativeness as part of social innovation, both at the firm and country level.

### 4.2. Implications of GII for Financial Innovation

In order to understand and assess financial innovative capacity at the firm level (BCG Innovation Study), we further provide an assessment of innovative capacity (technical and social) at the level of major countries of the world. For this purpose, we will use the data provided by the GII ranking.

We use a data set of the Global Innovation Index (GII), sub-indexes, and components for 2010 and 2020; in the following table, we summarize the main variables selected for the analysis. In our study, we chose to assess the relationship between GII and firm-level innovative capacity at 2020 vs. 2010 in order to have a broader picture (4 years if information offered by GII and GCI is cumulated, respectively, 2010 vs. 2009 and 2020 vs. 2019) of the factors/variables determining technical, social, and other types of innovation at the country/firm level. In the Table 10 we presented data description for this ranking (the name of each index is in bold).

**Table 10.** Data description.

| | Index/Component | Description |
|---|---|---|
| [1] | *Global Innovation Index* | Composite index calculated as the average between the average of the input and output sub-indices (7 pillars, 80 components) |
| [2] | *Innovation Input Sub-index* | Five input pillars capture elements of the national economy that enable innovative activities |
| [3] | *Innovation Output Sub-index* | The result of innovative activities within the economy (two pillars, with the same weight in the overall GII scores as the input sub-index) |
| [4] | *Gross expenditure on R&D (GERD)* | Total domestic intramural expenditure on R&D during a given period as a percentage of GDP |
| [5] | *Global R&D companies, average expenditure (top 4)* | Average expenditure on R&D of the top global companies |
| [6] | *ICT access* | A composite index that weighs five ICT indicators (20% each): fixed telephone subscriptions per 100 inhabitants; mobile cellular telephone subscriptions per 100 inhabitants; international Internet bandwidth (bit/s) per Internet user; percentage of households with a computer; and percentage of households with Internet access |
| [7] | *ICT use* | A composite index that weighs three ICT indicators (33% each): percentage of individuals using the Internet; fixed (wired)-broadband Internet subscriptions per 100 inhabitants; and active mobile broadband subscriptions per 100 inhabitants |
| [8] | *GERD performed by business enterprise* | Gross expenditure on R&D performed by business enterprise as a percentage of GDP |
| [9] | *GERD financed by business enterprise* | Gross expenditure on R&D financed by business enterprise as a percentage of total gross expenditure on R&D |
| [10] | *GERD financed abroad* | Percentage of gross expenditure on R&D financed abroad (billions, national currency) |
| [11] | *Joint venture/strategic alliance deals* | Joint ventures/strategic alliances: Number of deals and fractional counting (per billion PPP$ GDP) |
| [12] | *High-tech and medium high-tech output* | High-tech and medium-high-tech manufacturing (% of total manufacturing output) |
| [13] | *High-tech exports* | High-tech net exports (% of total trade) |
| [14] | *Computer and comm. service exports* | Computer, communication, and other service exports (% of commercial service exports) |
| [15] | *ICT service exports* | Telecommunication, computer, and information service exports (% of total trade) |
| [16] | *ICTs and business model creation* | Average answer to the question: in your country, to what extent do ICTs enable new business models? |
| [17] | *ICTs and organizational model creation* | Average answer to the question: in your country, to what extent do ICTs enable new organizational models (e.g., virtual teams, remote working, and telecommuting) within companies? |
| [18] | *Mobile app creation* | Global downloads of mobile apps, by origin of the headquarters of the developer/firm, scaled by PPP$ GDP (billions). |

In order to determine the paternity of evolution, i.e., the central tendency and the variability of the components, we used a quantitative analysis using descriptive statistics. In the first step, we identified global patterns of evolution for all countries included in the index calculation. In the second step, we selected countries that scored above the world average on the GII and determined the central tendency and variability for these groups of countries. For the group of countries that scored above the world average, we performed a correlation analysis.

Table 11 summarizes descriptive statistics for the GII and selected components in 2010, both for all countries included in the index calculation and for the group of countries scoring above the world average.

**Table 11.** Descriptive analysis of selected GII components (2010 scores).

| Index/ Component | Total Savings | | | | | | Above Mean Economies | | | | | |
|---|---|---|---|---|---|---|---|---|---|---|---|---|
| | Mean | Median | SD | Min | Max | Count | Mean | Median | SD | Min | Max | Count |
| [1] | 37 | 34 | 11 | 20 | 64 | 125 | 47 | 48 | 8 | 37 | 64 | 51 |
| [2] | 43 | 40 | 12 | 23 | 74 | 125 | 54 | 53 | 9 | 38 | 74 | 51 |
| [3] | 30 | 28 | 11 | 8 | 59 | 125 | 40 | 41 | 9 | 21 | 59 | 51 |
| [4] | 18 | 11 | 21 | 0 | 100 | 99 | 32 | 29 | 22 | 3 | 100 | 46 |
| [6] | 45 | 39 | 23 | 12 | 88 | 124 | 66 | 71 | 16 | 32 | 88 | 51 |
| [7] | 20 | 12 | 19 | 0 | 71 | 124 | 38 | 41 | 17 | 9 | 71 | 51 |
| [8] | 48 | 49 | 28 | 0 | 100 | 80 | 64 | 64 | 22 | 17 | 100 | 46 |
| [9] | 45 | 49 | 26 | 0 | 100 | 74 | 60 | 58 | 20 | 19 | 100 | 44 |
| [10] | 30 | 24 | 27 | 0 | 100 | 78 | 27 | 24 | 18 | 0 | 81 | 44 |
| [11] | 16 | 5 | 24 | 0 | 100 | 125 | 28 | 17 | 28 | 0 | 100 | 51 |
| [13] | 17 | 6 | 22 | 0 | 100 | 108 | 31 | 25 | 25 | 0 | 100 | 50 |
| [14] | 40 | 36 | 24 | 0 | 100 | 120 | 51 | 49 | 20 | 15 | 100 | 48 |
| [16] | 59 | 59 | 12 | 30 | 89 | 122 | 68 | 70 | 10 | 43 | 89 | 51 |
| [17] | 54 | 51 | 12 | 26 | 84 | 122 | 62 | 64 | 11 | 38 | 84 | 51 |

Overall, the trend in innovation performance (GII) in 2010 remains somewhat constant, with a relatively normal distribution of values. The best performers are countries such as Switzerland, Sweden, Singapore, Hong Kong (SAR), China, Finland, Denmark, the United States of America, Canada, Netherlands, the United Kingdom, Iceland, Germany, Ireland, Israel, New Zealand, Korea, Rep, Luxembourg, Norway, Austria, Japan, Australia, France, Estonia, Belgium, Hungary, Qatar, etc., while the lowest performers are associated with countries such as Senegal, Swaziland, Venezuela, Cameroon, Tanzania, Pakistan, Uganda, Mali, Malawi, Rwanda, Nicaragua, Cambodia, Bolivia, Madagascar, Zambia, Syrian Arab Republic, Tajikistan, Cote d'Ivoire, Benin, Zimbabwe, Burkina Faso, Ethiopia, Niger, Yemen, Sudan, Algeria, etc. As we will see later (Section 6, BCG ranking starting-point study), there are several European countries such as Sweden, Estonia, Belgium, Hungary, France, Finland, Denmark, etc., that are in good positions in the GII vs. GCI ranking, but do not have large/representative MNCs to include companies from these countries in the BCG ranking. How can this situation be explained? Also, in the sense mentioned, there are countries such as Qatar, Canada, New Zealand, etc., that do not have representative companies in the top tier of the innovation category that is summarized by the BCG study.

Among the selected components, the best scores were obtained in the infrastructure pillar on components such as ICT access and creative outputs (ICTs and business model creation, ICTs, and organizational model creation). In regards to business sophistication, notable performances were obtained regarding GERD performed by business enterprises (GERD financed by a business enterprise), but the importance of inter-firm cooperation through strategic alliances or joint ventures in innovation was rather low. The central trend towards the human capital and research pillar is also rather moderate, with a dispersed distribution of values across the board. The central tendency towards high-tech exports as a means of knowledge diffusion is also low, with a dispersed distribution of values.

In the group of countries performing above the world average in innovation, the central trend is relatively constant, but with a more dispersed distribution of variables. In this group, there is a noticeable trend towards a much stronger focus on infrastructure (through components such as ICT access and ICT use), business sophistication (GERD performed by a business and GERD financed by a business enterprise), and creative outputs (ICTs and business model creation; ICTs and organizational model creation). Also noteworthy is the increasing trend of knowledge diffusion, through high-tech exports and ICT service exports. However, the trend towards attracting external sources for research and development (GERD financed abroad) is less pronounced than the worldwide trend.

Table 12 illustrates the correlation matrix for the group of countries scoring above the world average in the GII, highlighting a number of significant influences that various components/variables have on innovation.

**Table 12.** Correlation matrix (above mean economies) 2010.

|  | [1] | [2] | [3] | [4] | [6] | [7] | [8] | [9] | [10] | [11] | [13] | [14] | [16] | [17] |
|---|---|---|---|---|---|---|---|---|---|---|---|---|---|---|
| [1] | 1.00 | | | | | | | | | | | | | |
| [2] | 0.92 | 1.00 | | | | | | | | | | | | |
| [3] | 0.91 | 0.70 | 1.00 | | | | | | | | | | | |
| [4] | 0.73 | 0.66 | 0.71 | 1.00 | | | | | | | | | | |
| [6] | 0.76 | 0.85 | 0.58 | 0.60 | 1.00 | | | | | | | | | |
| [7] | 0.79 | 0.87 | 0.61 | 0.69 | 0.90 | 1.00 | | | | | | | | |
| [8] | 0.66 | 0.64 | 0.58 | 0.74 | 0.47 | 0.61 | 1.00 | | | | | | | |
| [9] | 0.52 | 0.46 | 0.48 | 0.66 | 0.19 | 0.42 | 0.93 | 1.00 | | | | | | |
| [10] | −0.10 | −0.04 | −0.10 | −0.24 | 0.20 | −0.04 | −0.39 | −0.67 | 1.00 | | | | | |
| [11] | 0.53 | 0.67 | 0.31 | 0.30 | 0.42 | 0.45 | 0.41 | 0.41 | −0.31 | 1.00 | | | | |
| [13] | 0.40 | 0.30 | 0.39 | 0.24 | 0.02 | 0.14 | 0.48 | 0.50 | −0.28 | 0.08 | 1.00 | | | |
| [14] | 0.48 | 0.36 | 0.55 | 0.59 | 0.21 | 0.26 | 0.47 | 0.44 | −0.06 | 0.03 | 0.35 | 1.00 | | |
| [16] | 0.69 | 0.69 | 0.58 | 0.61 | 0.58 | 0.58 | 0.58 | 0.46 | −0.14 | 0.46 | 0.34 | 0.32 | 1.00 | |
| [17] | 0.71 | 0.70 | 0.60 | 0.61 | 0.55 | 0.53 | 0.56 | 0.47 | −0.15 | 0.55 | 0.30 | 0.35 | 0.95 | 1.00 |

In 2010, in the group of countries scoring above the world average, innovation performance is positively and significantly associated with gross expenditure on R&D, infrastructure (ICT access and ICT use), and business sophistication. Also, a significant influence has been had by the creation of business and organizational models through the use of ICT, which are positively associated with R&D financed and carried out by firms and with access to ICT (which predominantly means common social innovations of which some may later prove to be disruptive in society). An insignificant influence was exerted by R&D financed abroad. In regards to joint ventures and strategic alliances, there is a positive association with the use of ICT and R&D activity carried out within firms, positively influencing overall innovation performance. Thus, based on the 2010 GII data, it is quite clear that some factors/variables related to the innovative capacity of an entity (gross expenditure on R&D, with infrastructure (ICT access and ICT use) and with business sophistication, a business model, alliances and open innovation in R&D activity, etc.) are simultaneously found at the level of countries and/or firms that are considered to be innovative. However, it is extremely difficult to quantify and argue to what extent the existing technical or social innovative capacity in firms conditions/determines the same innovative capacity in countries A, B, C, etc. This is because, according to the Forbes rankings for both 2010 and more recently in 2022 [59], at the time of 2022, there were at least 2000 important/significant companies globally, yet the vast majority of them are far from making the BCG ranking, even if they also have quite good achievements on technical and social innovations. We analyzed 2010–2020 span time and there have been major changes in the number of companies and countries of origin that are included in the various international rankings (Forbes 2022; Fortune 2022; etc.). However, the dominance of US innovative capacity at the country and firm level remains; more recently, China and companies originating from this country have improved their international competitive

capacity. In a few cases, we find companies that have a turnover of USD 2–10 billion or more (Gold Fields—South Africa; Grifols—Spain; Cencosud—Chile; Dexus—Australia; Ayala Corporation—Philippines; Fertiglobe—United Arab Emirates; etc.) have a good position in Forbes 2022 and come from countries that do not have an above-average position in the GCI and GII. Simply put, the world's leading developed countries with relatively high GDP per capita dominate R&D activity (funding by firms as well as by the government) and perform better annually in terms of the number of patents, social innovations, financial innovations, etc.

Table 13 summarizes descriptive statistics for the GII and selected components in 2020, both for all countries included in the index calculation and for the group of countries scoring above the world average.

**Table 13.** Comparative descriptive analysis of selected components of the GII (2020 scores).

| Index/ Component | Total | | | | | | Above Mean | | | | | |
|---|---|---|---|---|---|---|---|---|---|---|---|---|
| | Mean | Median | SD | Min | Max | Count | Mean | Median | SD | Min | Max | Count |
| [1] | 34 | 31 | 12 | 14 | 66 | 131 | 47 | 46 | 9 | 34 | 66 | 53 |
| [2] | 43 | 41 | 12 | 20 | 70 | 131 | 55 | 55 | 9 | 40 | 70 | 53 |
| [3] | 24 | 21 | 13 | 7 | 63 | 131 | 38 | 36 | 9 | 23 | 63 | 53 |
| [4] | 17 | 10 | 20 | 0 | 100 | 131 | 33 | 27 | 22 | 3 | 100 | 53 |
| [5] | 20 | 0 | 32 | 0 | 100 | 131 | 45 | 49 | 35 | 0 | 100 | 53 |
| [6] | 61 | 66 | 20 | 21 | 93 | 131 | 77 | 80 | 11 | 38 | 93 | 53 |
| [7] | 54 | 54 | 24 | 3 | 90 | 131 | 73 | 77 | 13 | 25 | 90 | 53 |
| [8] | 10 | 2 | 18 | 0 | 100 | 131 | 24 | 18 | 21 | 0 | 100 | 53 |
| [9] | 32 | 23 | 30 | 0 | 100 | 131 | 60 | 61 | 22 | 0 | 100 | 53 |
| [10] | 13 | 3 | 20 | 0 | 100 | 131 | 26 | 23 | 24 | 0 | 100 | 53 |
| [11] | 17 | 7 | 23 | 0 | 100 | 131 | 31 | 18 | 29 | 2 | 100 | 53 |
| [12] | 25 | 17 | 24 | 0 | 100 | 131 | 43 | 43 | 22 | 3 | 100 | 53 |
| [13] | 29 | 20 | 27 | 0 | 100 | 131 | 51 | 51 | 25 | 3 | 100 | 53 |
| [15] | 21 | 16 | 21 | 0 | 100 | 131 | 30 | 23 | 26 | 1 | 100 | 53 |
| [17] | 53 | 53 | 17 | 0 | 84 | 131 | 66 | 65 | 11 | 44 | 84 | 53 |
| [18] | 13 | 2 | 23 | 0 | 100 | 131 | 27 | 16 | 28 | 0 | 100 | 53 |

Source: Processed from GII2020.

The trend for the Global Innovation Index in 2020 remains somewhat constant, close to the trend recorded in 2010 in terms of value distribution. The best performers are Switzerland, Sweden, the United States of America, the United Kingdom, the Netherlands, Denmark, Finland, Singapore, Germany, the Republic of Korea, Hong Kong, China, France, Israel, China, Ireland, Japan, Canada, Luxembourg, Austria, Norway, Iceland, Belgium, Australia, Estonia, the Czech Republic, New Zealand, Malta, Cyprus, Italy, Spain, and Portugal, while the lowest performances are associated with countries such as Bolivia, Guatemala, Ghana, Pakistan, Tajikistan, Cambodia, Malawi, Côte d'Ivoire, Lao People's Democratic Republic, Uganda, Bangladesh, Madagascar, Nigeria, Burkina Faso, Cameroon, Zimbabwe, Algeria, Zambia, Mali, Mozambique, Togo, Benin, Ethiopia, Niger, Myanmar, Guinea, Yemen, etc. Of the selected components, the best scores were obtained in the *infrastructure* pillar (on components such as ICT access and ICT use), *business sophistication* (GERD financed by a business enterprise), and *creative outputs* (ICTs and business model creation; ICTs and organizational model creation), while the central trend towards the *human capital and research* pillar is still rather moderate, with a dispersed distribution of values across all situations. Also notable is the central trend of increasing high-tech exports, as a means of knowledge diffusion, but still with a dispersed distribution of values; this time, among the leaders are Malaysia, Vietnam, the Philippines, the Republic of Korea, China, and Singapore, together with the Czech Republic, France, Germany, etc. Together with other conclusions that can be drawn, it follows that the GCI and GII rankings are still largely dominated by the developed countries of the world by 2020. In a few cases, we will find, in 2020, companies that are globally significant [41,42] but come from countries that are well below the average GII ranking. Therefore, we see that there is a

certain conditionality/correlation between existing technical and social capacity at the firm/company level and innovative capacity and/or competitive position at the country level. In the next part of the study (Section 5), we address the same topic, but from the microeconomic perspective of analyzing financial innovations as part of social innovations at the firm/country level.

In the group of countries performing above the world average in GII (Table 14), the central trend is relatively constant, but with a more dispersed distribution of variables. In this group, there is a perceptible trend towards a much stronger focus on infrastructure (through components such as ICT access and ICT use), *business sophistication* (GERD financed by a business enterprise), and *creative outputs* (ICTs and business model creation; ICTs and organizational model creation); at the same time, the central trend towards the *human capital and research* pillar, although still moderate and rather spread, is more consistent than the global trend. The orientation towards attracting external sources for research and development (GERD financed abroad) should also be highlighted, with Israel, the Czech Republic, Austria, Iceland, Belgium, Sweden, Finland, etc., leading the way. Japan, Malaysia, China, Thailand, India, etc., have shown less interest in this direction. Last but not least, mobile app creation is showing a more consistent, albeit still dispersed, upward trend than the global trend.

**Table 14.** Correlation matrix 2020 (economies with above-mean scores).

| | [1] | [2] | [3] | [4] | [5] | [6] | [7] | [8] | [9] | [10] | [11] | [12] | [13] | [15] | [17] | [18] |
|---|---|---|---|---|---|---|---|---|---|---|---|---|---|---|---|---|
| [1] | 1 | | | | | | | | | | | | | | | |
| [2] | 0.95 | 1.00 | | | | | | | | | | | | | | |
| [3] | 0.95 | 0.81 | 1.00 | | | | | | | | | | | | | |
| [4] | 0.75 | 0.74 | 0.68 | 1.00 | | | | | | | | | | | | |
| [5] | 0.69 | 0.68 | 0.64 | 0.70 | 1.00 | | | | | | | | | | | |
| [6] | 0.60 | 0.68 | 0.47 | 0.41 | 0.31 | 1.00 | | | | | | | | | | |
| [7] | 0.74 | 0.79 | 0.62 | 0.54 | 0.37 | 0.85 | 1.00 | | | | | | | | | |
| [8] | 0.70 | 0.67 | 0.66 | 0.98 | 0.66 | 0.35 | 0.47 | 1.00 | | | | | | | | |
| [9] | 0.43 | 0.37 | 0.45 | 0.44 | 0.46 | 0.15 | 0.30 | 0.48 | 1.00 | | | | | | | |
| [10] | 0.43 | 0.38 | 0.43 | 0.55 | 0.18 | 0.28 | 0.35 | 0.54 | −0.02 | 1.00 | | | | | | |
| [11] | 0.60 | 0.63 | 0.52 | 0.30 | 0.24 | 0.48 | 0.56 | 0.27 | 0.02 | 0.23 | 1.00 | | | | | |
| [12] | 0.47 | 0.39 | 0.50 | 0.50 | 0.47 | −0.01 | 0.11 | 0.53 | 0.51 | 0.24 | −0.04 | 1.00 | | | | |
| [13] | 0.24 | 0.11 | 0.34 | 0.34 | 0.17 | −0.23 | −0.04 | 0.39 | 0.46 | 0.18 | −0.18 | 0.74 | 1.00 | | | |
| [15] | 0.07 | −0.03 | 0.15 | 0.04 | 0.04 | −0.20 | −0.18 | 0.08 | −0.24 | 0.33 | 0.22 | 0.05 | −0.04 | 1.00 | | |
| [17] | 0.81 | 0.81 | 0.73 | 0.60 | 0.53 | 0.45 | 0.61 | 0.53 | 0.28 | 0.42 | 0.58 | 0.34 | 0.24 | 0.06 | 1.00 | |
| [18] | 0.37 | 0.35 | 0.35 | 0.25 | −0.01 | 0.21 | 0.29 | 0.25 | 0.03 | 0.27 | 0.49 | 0.02 | −0.02 | 0.39 | 0.29 | 1.00 |

In 2020, the innovation performance of the 53 countries scoring above the world average was positively and significantly associated mainly with resources invested in R&D activity, infrastructure, R&D activity carried out by firms, and especially the creation of new organizational models under the impact of ICT (the United States of America, Sweden, Finland, the Netherlands, Estonia, the United Kingdom, Denmark, Germany, Switzerland, Norway, Israel, Canada, Iceland, Singapore, Luxembourg, and Belgium). The creation of these new organizational models is positively and significantly associated with ICT use, cooperation through joint ventures and strategic alliances, and R&D activities carried out by firms. The conclusions that can be drawn from the GCI and GII rankings over the last 10 years largely confirm Porter's concept of "the five forces" shaping competition at the level of industrial sectors and/or countries [60].

In contrast to 2010, the role of strategic alliances, but especially R&D financed abroad, has increased. High-tech and medium-tech production is mainly supported by in-house R&D, and high-tech exports are positively associated with R&D financed by firms. Moreover, in 2020, the role, albeit still modest, of mobile technologies is being felt, which is positively associated with cooperation through JVs and strategic alliances.

Up to this point, our assessments have predominantly focused on the macro-economic perspective of country/firm-level innovation processes through the use of disruptive technologies. In the following sections of the study, we aim to focus the analysis on the same topic from a microeconomic perspective, i.e., with reference to the realities in different categories of firms (especially MNCs). In the innovative sense, we then aim to "intersect" the two perspectives of analysis in order to reach some clearer conclusions on the role/importance of disruptive/digital technologies in the case of financial innovations made by firms.

## 5. Financial Innovations with Disruptive Technology

As we argue in our study, there is no clear/unambiguous distinction between "financial innovation" and other types of "social innovation" in the entire international literature. Previously, we listed some of the world's leading countries that are significantly above the GII average at both 2010 (Table 11) and 2020 (Table 13). Comparing the information provided by the GCI and the GII (for about a decade of the analysis of the two rankings), it can be deduced that the world's leading countries with higher annual nominal GDP per capita will also have a better position with regard to innovative capacity in general. Innovative capacity in such leading countries of the world is, however, determined with a complex of factors (values, institutions, government policies, government- and/or firm-funded R&D investments, university education, competition between firms, scientific competition, etc.), even for smaller countries (Sweden, Denmark, Finland, Spain, Portugal, Italy, etc.) that do not have MNCs to be included in the BCG ranking of the most innovative companies. Some important questions arise that are related to the basic idea of our study: "In all or most of these countries, do we find MNCs and SMEs that are technically, socially, and financially innovative? Do such firms excel or not on all three innovation dimensions? Do such firms excel or not on all four types of innovation, i.e., product, process, marketing, and organization?". It is difficult to formulate clear and reasoned answers to such questions because, globally, there are about 2000 companies (Fortune 2000–2022) that are in various rankings and have a significant innovative capacity, even if very differently from one organization to another. In our study, we included in the analysis only 50 companies considered to be the most innovative from 2004–2022, according to the BCG (Boston Consulting Group) study, and they are analyzed together in Table S2. Also in the sense mentioned, we admit that there are relatively clear/consistent statistical data for the innovative capacity of the main countries of the world (studies such as GII, Global Competitiveness Report, WIPO [61], etc.), but there are only a few studies on the same subject on MNCs coming from different countries as the location of headquarters.

Over the past three decades, disruptive technologies have emerged as a common element/vector for knowledge acquisition, exploitation, and transformation into patents and various types of social innovation [62,63] both at the level of firms, other organizations, and countries. Any firm and/or other type of organization, regardless of size or country of origin, has the opportunity to use digital and other disruptive technologies (robots, satellites, biotechnologies, etc.) to improve its organizational, production, and market processes.

Any firm can be approached as a socio-economic system, i.e., a hub that is in permanent connection with certain stakeholders (investors, shareholders, managers, suppliers, employees, customers, banks, insurers, state institutions, universities, other firms, etc.). Current relations with the vast majority of stakeholders involve financial relations (i.e., with investors, shareholders, managers, suppliers, employees, customers, banks, etc.). Any firm, regardless of size, needs to manage capital and cash flows extremely preventively, in order not to end up making "book profit" and at the same time being in a financial crisis [64]. These ongoing relationships with various other entities require the widest possible use of digital and other types of technology, and at the same time require financial innovations as part of social innovations [24]. In Figure 5, we present a firm approached as a social–economic system in its relations with various stakeholders.

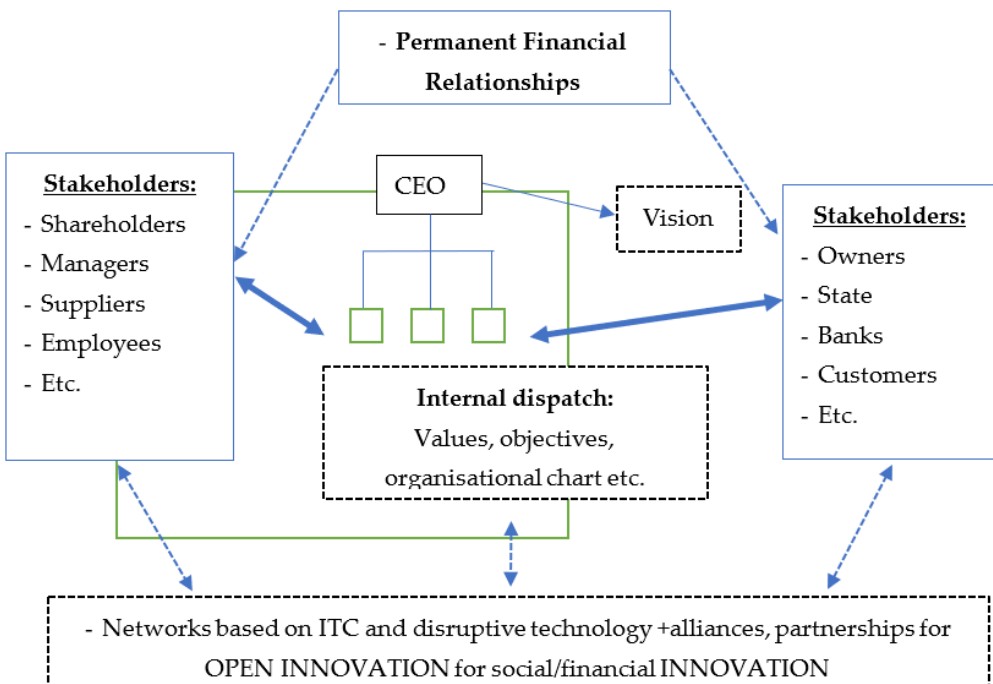

**Figure 5.** A company approached as a "system" and financial relationships with some stakeholders. Sources: author's design.

The main idea that we want to highlight through the graphic sketch in the following figure is that any firm/company, from the moment of its establishment, inevitably develops multiple financial relationships, along with commercial/human relationships with different stakeholders. According to the UNCTAD classification [51], companies are classified into non-financial companies (those in various industries and some KIS sectors) and financial companies, i.e., banks, insurance companies, investment funds, etc. This does not mean, however, that companies in the non-financial category would not engage in various financial relationships with numerous stakeholders. In fact, the revolution brought by fintech has primarily manifested itself in various financial markets and all assessments show that the fintech industry has become of utmost interest to non-financial companies as well (refs. [14,31,35]; etc.). The study we include in Table S2 on the BCG ranking only includes non-financial companies in the analysis, but the Forbes study also includes companies such as large banks and insurance companies. Therefore, the reality encountered in the global economy at both the MNCs and SMEs level is extremely diverse and a summary on fintech industries/markets [52] is useful in the context of the rapid advancement of disruptive technologies in different segments of international financial markets. As we show later, also in this section of the study, disruptive strategies or alliances recently implemented by large MNCs can be considered as disruptive innovations and will have major financial effects in the coming years even in traditional industries such as in the global automotive industry. Also, in the sense invoked, how a traditional MNC or fintech start-up prudently manages its own cashflow has become a key issue for imposing financial innovations as part of social innovations. There are therefore several direct/indirect arguments to clearly highlight the position of the socio-economic system that any business organization has and the fact that this system is essentially different from a national economy, as argued by Krugman [15].

In so far as we approach a firm as an open socio-economic system, depending on the field in which it is located (manufacturing industry, knowledge intensive services, and other fields), it follows that a large part of the relationships it engages in with other entities also involve and/or require financial flows. Since a large part of the relationships that any firm (MNCs and SMEs), in any country, engages in materialize as financial relationships, it can be considered empirically, according to Pareto's principle, that 80% of the social innovations

that a firm/organization is able to achieve annually are in fact financial innovations (we have previously shown that applying the Pareto principle only allows us to suggest the direction in which the answer to the question of how to evaluate financial innovations should be identified and that in many cases, this principle will have corresponding weights of 50–50%, 60–40%, etc.). Therefore, any technological input that a firm uses to improve its innovative capacity, both technical and social, is highly beneficial and is reflected in the financial innovations achieved annually by firms. This is even though we do not have studies measuring/quantifying a firm's financial innovative capacity as part of its social innovative capacity. In turbulent/chaotic times, argues the author of [65], the CEO of any company should consider that "financial strength" is far more important than revenue and profit; this was true in the 1980s and has become much more evident after 2008 and up to the present.

As can be seen from Table S2 and the tables showing the companies in the BCG study (Section 6 of the study), Toyota was ranked 21st in the 2022 ranking, but is no longer in the 2023 ranking (we return to this sample of companies in the last part of the study). Toyota's cash-flow management strategy in relation to the thousands of stakeholders with whom it works on a regular basis has been extremely conservative over the past six decades; the conservative nature of this strategy has enabled the company to more easily overcome the crisis of 2008–2010 [66]. More recently, starting in 2015, Toyota adopted a disruptive strategy of exploiting EV (electric vehicle) patents in the sense that it offers around 23,000 patents without paying annual royalties to any other competitor [67,68]. A few years later, Tesla adopted a disruptive strategy similar to Toyota for EVs manufactured by it; the EV car industry has clearly become a disruptive industry and one that relies heavily on the use of digital technologies. Also relatively recently, in 2023, Ford, Tesla, GM, and Rivian formed an alliance to jointly operate and improve EV battery charging equipment; the alliance invoked is clearly also disruptive in nature and will generate major financial implications for the four members, but also with reference to the use of financial technologies to support social innovations. Questions arise that are tangential to the basic idea of the present study: "What are the financial and open innovation implications of the EV strategies already applied by Toyota and Tesla?"; "What are the financial implications in relation to the thousands of stakeholders of the alliance formed by the four US companies?"; "How will such realities in the world of large non-financial MNCs reflect on the evolution of the fintech concept?". We are not in a position to answer and it is not the purpose of this study to formulate answers to questions of the type mentioned. However, it is easy to deduce that even large MNCs in various industries are able to exploit digital technologies intensively as part of various disruptive technologies to achieve new financial innovations, some of which will prove to be disruptive over time. As we will show later (point 6.3 in Section 6 of the study), a large part of the companies considered innovative by BCG have been applying innovative strategies and practices on cash flow management and other financial aspects in their relationship with most of their stakeholders, since 2–3 decades ago (as shown in Figure 5). Also, in the sense invoked, digital and other types of disruptive technologies have supported established companies (financial and non-financial) to form financial alliances, strategies, and practices of various types to realize "open innovation" networks [38] since two decades ago already, i.e., before the individualization of fintech markets.

We mentioned earlier (Section 4.1) about the four types of innovation according to the existing OECD [50] classification (product, process, marketing, and organizational). The use of digital/disruptive technologies supports a firm's effort to bring elements of novelty in any of the four directions/types of innovation. However, some differentiations can be made by categories of firms, depending on the domain in which they are located (but not on the countries they belong to), respectively:

- Firms in high-tech and medium-high-tech sectors are more likely to achieve technical, product, and process/technological innovations/inventions; the field does not limit innovation in these entities on marketing and organizational issues;

- Firms in medium-low-tech and low-tech sectors are more likely to achieve social innovation, i.e., on marketing and organizational issues; the area in which they are located partially limits the achievement of product and process innovation (but there may be many exceptions to this rule);
- Firms in various service sectors are more likely to achieve social innovations, i.e., on marketing and organizational issues; also, the area in which they are located partially limits the achievement of product and process innovations (but there may be many exceptions to this rule).

## 6. Main Findings

### 6.1. Driving Forces for Financial Innovation

From a historical perspective, social innovations such as the newspaper, the insurance system, hire–purchase, labor relations, or the pension system have fundamentally changed the Western economy/society, as Drucker points out [21]. It is obvious, however, that social innovations of any kind cannot be protected with patents, as is the case with a technical innovation (only some social and/or technical innovations can be protected). In Table S1, we summarized together the technical innovative capacity given with the number of patents of the 50 companies in the BCG study [58], and the social innovative capacity for the same companies (reflected by the number of trademarks, industrial designs, etc., registered with various state agencies, as appropriate). Some of the social innovations, i.e., trademarks, trade names, industrial designs, logos, symbols, etc., can be protected by firms through registrations with the agencies under which patents are usually granted (national agencies such as USPTO in the case of the US, EPO in the case of Europe, etc.). Even in such cases where the law offers some protection for any element of novelty brought by a firm, some social innovations will become sources of orientation for competitors and will gradually be taken over by other firms. Only in the case of social innovations made by a firm on the basis of employees 'tacit knowledge, which is dependent on the values they believe in and is more difficult to transform into explicit knowledge (i.e., know-how, ways of solving a problem, mental models, etc.), the organization can hope for a longer protection of such elements of novelty.

There are, as we argue below, at least three driving forces that directly support any firm to achieve social innovations in general (only some of which will also prove to be disruptive social innovations). As the case may be, it is likely that the result of the cumulative action of the three driving forces will be a mix/combination of technical innovation and social innovation. In other words, only each firm can correctly assess on which direction of action it should focus its resources in order to overcome the specific crisis of the period 2021–2022 through social innovation. As we have shown above, most of the social innovations made by firms will be found in the market and/or in society as financial innovations.

The three types of driving forces for financial/social innovation are the following:

✓ Firstly, the technologies already existing in society, especially digital technologies (but not the only ones), are a real catalyst for companies to maintain their position in the market, improve performance, etc. This means massive recourse to digital technologies, which requires skilled employees, investments, own intranet systems, permanent connections with customers/suppliers and other stakeholders, etc. This trend of an increasing use of digital technologies started in the 1990s, became more pronounced in the context of the Great Recession of 2008, and has become essential for the survival of firms from 2020 onwards. Computer networks, communication networks, and satellites have made e-commerce possible, which in turn has fundamentally changed social and business relations in modern society. In the age of e-commerce, Drucker argues, even small, locally operating firms must be managed transnationally if they are to survive [62]. The new IT&C systems that firms are developing are leading to changes in processes within a firm and in its relationship with the outside world, changes that significantly alter the organizational culture. Almost every employee in firms and other organizations in modern society can gradually make small improvements in the

performance of job tasks when using elements of digital technology. In other words, digital/disruptive technology supports employees at all levels of an organization chart to contribute directly to the realization of social innovations by firms. For some of the social/financial innovations achieved to be disruptive, there needs to be a 'mindset' across the culture of an organization that aims to systematically take advantage of the benefits given with digital technologies [23]. There is no known 'recipe' or mechanism with which firms should proceed to achieve disruptive social/financial innovation. Each firm needs to define and identify its own direction in which it should focus, to act systematically to understand the constraints and opportunities given with the market on the product/service offered. Only then, after understanding what has real value for its customers [64], can a firm choose a favorable combination of existing technologies that will support its continuous innovation effort.

✓ Another driver forcing companies to resort to technical and social innovations is the competition in the market itself. Simply put, the market, with all its imperfections, offers constraints as well as opportunities for any business organization. Particularly in the context of a global recession such as the current one, it is essential for companies to understand in depth what is of value to the customers to whom the product/service offered is addressed. More importantly, the very values to which customers (end consumers) relate are changing rapidly in a global and increasingly interdependent society. The values, preferences, consumption habits, time-sharing of each individual, and social conditioning imposed by governments are all changing in a social climate defined by uncertainties. What directions for action can firms see in this new social climate?

So, for the whole period after 2008 and up to the present, the market and what the customer considers to be of value as a product/service has become the key element for companies to relate to. The general allocation of funds to R&D and obtaining patents and various trademarks, designs, etc., must be "doubled" by a systematic CEO effort to make disruptive innovations and especially disruptive financial innovations. The ownership of a large portfolio of patents by leading companies in high-tech and medium-tech sectors seems to be no longer sufficient to overcome the current global recession. The question arises as to whether some of the patents held by such companies will be transformed into low-end disruptive products/services. For companies in low-tech and service industries, the use of digital technologies and investment in their own IT&C systems seems to be the best way to adapt the product/service to the new requirements of low-end disruptive markets.

✓ Thirdly, managers and particularly the CEO and their team have become, perhaps more than a decade ago, the essential vector or "driving force" for firms (in any economic sector) that have a distinct strategy for achieving disruptive technical and social innovation. This CEO strategy/vision also requires skilled employees who are willing to continuously learn and to accumulate new explicit and tacit knowledge. In fact, the role of professional managers became essential with the emergence of large corporations, first in the US from 1880 to the present. The emergence of managerial capitalism was an economic phenomenon of American origin that then spread to Europe and Japan [69]. During the post-war period, the importance and role of professional managers has increased in all major countries of the world (including China in the last four decades). Since the 1980s, as digital technologies have become ubiquitous in society, the tasks to be performed by CEOs and the skills required have increased greatly in complexity.

In Figure 6 below, we present the three "driving forces" that together support the systematic effort of firms to achieve social innovation, of which the vast majority (about up to 80%) will be financial.

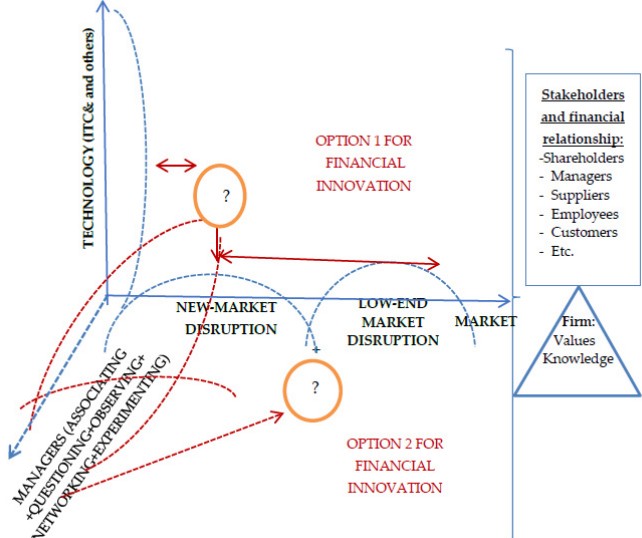

**Figure 6.** The driving forces for financial/social innovation. Sources: Author's design.

In relation to the three "driving forces" suggested by us in Figure 6, two or more variants (marked by us with a question mark in the figure) emerge, quite obviously, as a solution of connection/intersection between the support given by technologies and what the market considers to be of value to consumers. As can be easily deduced from the figure, the first variant of "intersection" or connection, usually accessible, can be more easily intuited by managers in established companies in a given market. As it can be seen, "option 1 for financial innovation" is given with the combination of the benefits brought with technologies and new market disruption, as an immediate direction of action for the managers of firms; only in the second place, the mix of the two elements can also be carried out in relation to "low-end market disruption". The financial relationships that a firm engages in with various categories of stakeholders (argued in Section 4 of the study; Figure 3), together with the values, knowledge, and training of the firm's human resources, will determine over time the financial/social innovation capacity of the organization. Through the policies applied, investments in R&D, the quality of education, and the quality of the infrastructure built, governments can support this direction of action in innovative firms (even if no distinction is made between technical and social innovation through such policies). On the other hand, when the support given with ICT would intersect with the value given with the market in any other location in the three-dimensional plane (of analysis), it becomes much more difficult for any manager to intuit at what point the "intersection" between the two might take place. As noted from the figure, along this line of action and/or intuitive assessment of social innovation opportunities, managers should first build their KM/innovation strategy on the basis of going through steps such as associating, questioning, observing, networking, and experimenting [32]. Therefore, a different vision, thinking differently, is needed for managers to be able to intuit what would be "option 2 for financial innovation" action to achieve social innovations of which most of them will turn out to be financial innovations. In this case, depending on the vision and intuition of managers, any kind of intersection between technologies and the two types of markets becomes possible, which then confirms or disproves the disruptive nature of any innovation (be it social, financial, or other). It is not the size of a firm that limits an organization's innovative capacity, Drucker argues, but the kind of developed culture and entrepreneurship that can be learned over time through practice [64].

### 6.2. Financial Innovation of MNCs and SMEs

Our assessments of financial innovation globally to this point in the study lead us to conclude that any type of organization, i.e., MNCs, SMEs, universities, public institutions, etc., can achieve significant, modest, or more significant social innovations through the

extensive use of digital/disruptive technologies. This means that any of the tens of millions of firms globally can have an effective KM, HR, and continuous innovation strategy. There are no studies that highlight/synthesize the technical and social innovative capacity for millions of firms in Europe, America, Asia, and other regions of the world. Specifically in the case of our study, we restrict ourselves to a more in-depth analysis of innovative capacity at the level of 50 major companies that have been monitored by BCG over the last two decades (approximately).

The BCG study is based on a questionnaire methodology that is applied annually to top executives of leading companies in both industries and services globally (the questionnaire is structured on four dimensions: Global Mindshare; Industry Peer View; Industry Disruption; and Value Creation). The very structure of the BCG questionnaire, used to determine each year which are the most innovative companies in the world, takes into account the strategies and/or disruptive innovations achieved by the companies, as well as the practices and platforms used by innovative companies. The BCG methodology lists three pillars for practices (portfolio management, funnel management, and project management) and eight pillars for platforms (idea to impact process, talent and culture, organization setup and ecosystems, performance management and metrics, innovation governance, innovation domains, and innovation ambition). Some of the 50 companies (depending on the score obtained on each pillar) will be considered as more committed innovators and others as less committed to innovation (as an explicit top management strategy). In the same sense, we find that the content of the 11 pillars of the BGC methodology is only partially matched by the content of the various pillars of the GCI and GII rankings (the application of KM and systematic innovation is to a large extent dependent on a CEO's vision, the company culture, the cyclical evolution of the business, and other similar factors; only to a certain extent can the innovative capacity of the country of origin influence/determine the innovation orientation of the companies). In Table 15, we present the BCG ranking for 2022, and in Table 16, we present the BCG ranking for 2023; in our substantive analysis, we will only deepen the ranking for 2022 for which we have the extended assessment in Table S2 of the study.

**Table 15.** The Most Innovative Companies of 2022.

| Ranking | | | | |
|---|---|---|---|---|
| **1–10** | **11–20** | **21–30** | **31–40** | **41–50** |
| 1 Apple | 11 Meta [1] | 21 Toyota | 31 Xiaomi | 41 Tencent |
| 2 Microsoft | 12 Nike | 22 Alibaba | 32 eBay | 42 General Motors |
| 3 Amazon | 13 Walmart | 23 HP | 33 Hyundai | 43 Ford 🟢 |
| 4 Alphabet | 14 Dell | 24 Lenovo | 34 Procter | 44 Intel 🟢 |
| 5 Tesla | 15 Nvidia ⬤ | 25 Zalando ⬤ | 35 Adidas | 45 ByteDance ⬤ |
| 6 Samsung | 16 LG | 26 Bosch | 36 Coca-Cola | 46 Panasonic ⬤ |
| 7 Modern | 17 Target | 27 Johnson & Johnson | 37 3 M 🟢 | 47 Philips |
| 8 Huawei | 18 Pfizer | 28 Cisco | 38 PepsiCo | 48 Mitsubishi |
| 9 Sony | 19 Oracle | 29 General Electric | 39 Hitachi 🟢 | 49 Nestlé 🟢 |
| 10 IBM | 20 Siemens | 30 Jingdong 🟢 | 40 SAP | 50 Unilever 🟢 |
| | | New entrant ⬤ Returned 🟢 | | |

Source: BCG Most Innovative Companies (MIC) Report 2022; the full version of the ranking and our assessment can be found in Table S1. Note: Industry classification is based on Capital IQ; some companies play across industries. [1] Facebook became Meta in 2021; in previous BCG Most Innovative Companies reports, it appeared under the name Facebook.

**Table 16.** The Most Innovative Companies of 2023.

| Ranking | | | | |
|---|---|---|---|---|
| **1–10** | **11–20** | **21–30** | **31–40** | **41–50** |
| 1 Apple | 11 Pfizer (+7) | 21 Rock | 31 Sony | **41 Saudi Aramco** |
| 2 Tesla (+3) | 12 Johnson & Johnson (+15) | 22 Oracle (−3) | 32 Sinopec | 42 Coca-Cola (−6) |
| 3 Amazon | 13 SpaceX | 23 BioNTech | 33 Hitachi (+6) | 43 Mercedes-Benz Group [1] |
| 4 Alphabet | 14 Nvidia (+1) | 24 Shel | 34 McDonald's | 44 Alibaba (−22) |
| 5 Microsoft (−3) | 15 ExxonMobil | 25 Schneider Electric | 35 Merck | 45 Walmart (−32) |
| 6 Modern (+1) | 16 Goal (−5) | 26 P&G (+8) | 36 ByteDance | 46 PetroChina |
| 7 Samsung (−1) | 17 Nike (−5) | 27 Nestlé (+22) | 37 Bosch (−11) | 47 NTT |
| 8 Huawei | 18 IBM (−8) | 28 General Electric (+1) | 38 Dell (−24) | **48 Lenovo (−24)** |
| 9 BYD Company | 19 3 M (+18) | **29 Xiaomi (+2)** | 39 Glencore | **49 BMW** |
| 10 Siemens (+10) | 20 Tata Group | **30 Honeywell** | **40 Stripe** | **50 Unilever** |
| | | xxx—Returned xxx—New entrant | | |

Sources: BCG [70] Global Innovation Survey 2023; BCG analysis. Note: +/− indicates change from 2022 MIC ranking. [1] Mercedes-Benz Group was previously identified as Daimler; in bold are new entry.

The same BCG study of the most innovative companies globally for 2023 is shown in Table 16:

The comparative analysis of Tables 15 and 16 shows that there is a very robust competition between internationally known MNCs and that there are annual changes in positions (companies dropping out of the ranking and coming in; changes in the actual position of those remaining; etc.). In 2023, companies such as Saudi Aramco or PetroChina have recently entered (so companies from countries with a more modest position in the GII can enter this ranking; in the same sense, although it is a developing country by GDP per capita, it has improved its competitive position in the last decade, and an increasing number of companies are starting to find themselves in various international rankings).

In the manufacturing industry, there is a separation of industries into four broad sectors: high technology, medium-high technology, medium-low technology, and low technology [71,72]. In regards to the services sector, depending on the intensity of technology use, the knowledge-intensive services (KIS) sector has a special place; in this sector, we include high-tech knowledge-intensive services (HTKIS); knowledge-intensive market services (KIMS); knowledge-intensive financial services (KIFS); and other knowledge-intensive services (OKIS) [72]. In the category of other services, which are less knowledge-intensive, we will classify areas/sectors such as tourism, road transport, retail, etc.; they are classified under the Eurostat less-knowledge-intensive services (LKIS) [72].

Based on the data in Tables 15 and 16 and Table S2, we present below a more in-depth analysis of the situation of the 50 companies in the BCG 2022 ranking. This analysis highlights the following issues:

It is possible to perform a pairwise comparison between variables to identify the type of link that might exist between the 50 companies (based on geographical location, the number of employees, net income, registered trademarks, and patents obtained by each entity, as summarized in Table S2). In Figure 7, we present the summary of the statistical evaluation of this comparison between the 50 companies.

In Figure 7, we retained only four variables for pairwise comparisons, since the variable continents are clarified in Table S2 and maintained in the first part of the study (the structure by continents is the following: 27 firms—USA; 15 firms—Asia; and 8 firms—Europe). The retreat of the figure only by including the other four variables leads us to the conclusion that the 50 firms have a fairly wide distribution on the basis of the characteristics taken

into account (the synthetic evaluation shows aspects such as the following: most firms have up to 500,000 employees, obtain up to 2500 patents per year, have up to 1500 social innovations in their portfolios, achieve up to a 40 billion net income per year, etc.). The same preliminary conclusions can be drawn from the empirical evaluation of the data in Table S2. The data in Table S2 show a rather large dispersion of the variables taken into account (examples: from about 400 employees at Moderna to about 420 thousand employees at Bosch; from a symbolic profit of 0.1 billion at Zalando to a profit of almost 100 billion USD at Apple; from 40 social innovations at Zalando to over 1500 social innovations at other companies; and from 10 patents per year at Moderna to over 6000 patents per year at Samsung), which is reflected in the distribution of companies in the previous figure.

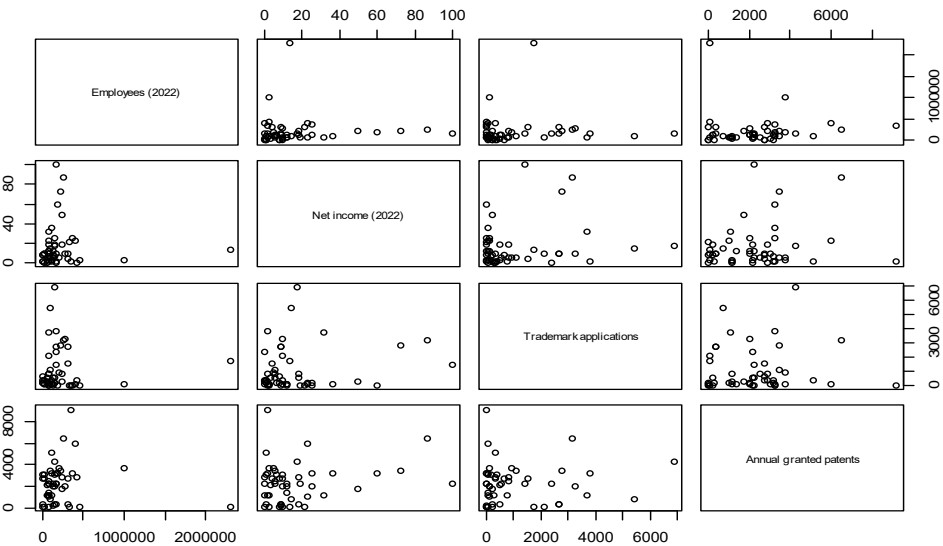

**Figure 7.** Pairwise comparisons between variables to identify the link type.

If we select only the number of patents (as equivalent for technical innovations) vs. the number of trademarks, trade names, logos, etc. (as equivalent for social innovations), the distribution of the 50 firms is shown in Figure 8.

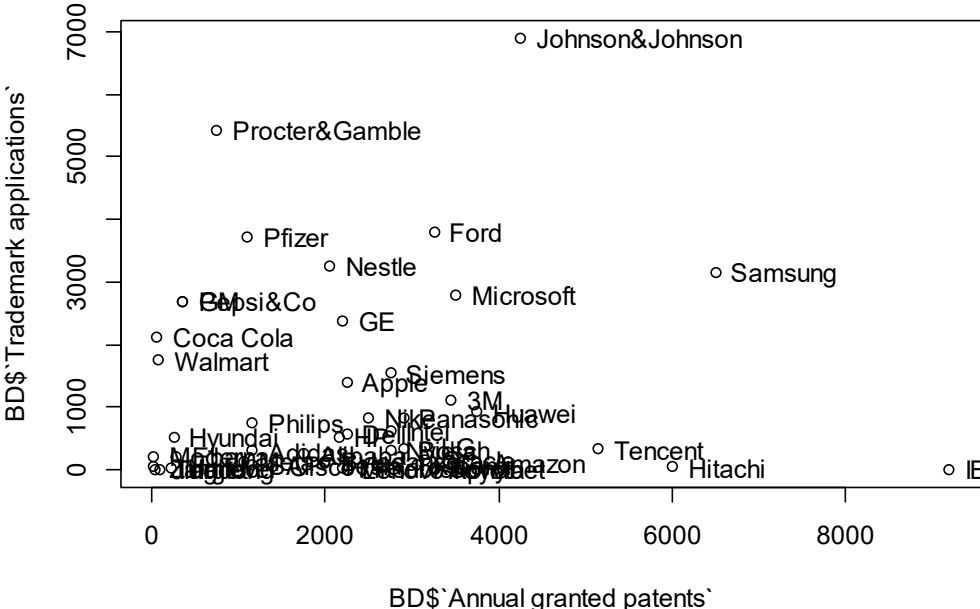

**Figure 8.** Analysis of the relationship between the variables: number of patents (technical innovations) and number of registered trademarks (social innovations).

From what has been presented up to this point regarding the distribution of the 50 companies in the BCG ranking, only a few preliminary conclusions can be formulated with respect to the objective of our research. We recall the basic idea of the present study, i.e., to further analyze the relationship between GCI, GII, and disruptive technology simultaneously at the level of core countries and MNCs considered to be innovative and successful. The question included in the title of the study "How do we evaluate financial innovation made by firms?" is predominantly rhetorical and aimed only at suggesting some directions/clarifications regarding the concept of fintech and innovations made by firms using digital technologies. The synthesis presented by us previously (Section 6.1 of the study) allows us to mention that disruptive technologies have now become a kind of "driving force" for financial/social innovation by firms. The rather in-depth assessment of the realities and prospects that lie ahead for the 50 companies in the BCG study (the present section of the study) gives us an indirect confirmation of the role that disruptive financial technologies are playing in various segments of international financial markets (particularly those where fintech start-ups are already operating). On the basis of the BCG ranking for 2022, also taking into account the analysis based on the GCI (Section 4.1) and the GII (Section 4.2), we find that there are some common "driving forces" to explain the innovative capacity both at the firm and country level. This is because some of the pillars of the BCG methodology (e.g., Global Mindshare, Value Creation, etc.), as well as most of the eight elements that form the basis of the strategies applied by innovative firms (practices and platforms), have as their country-level equivalent education, training, stakeholder relations, government policies, and other variables that explain innovative capacity from a macro-economic perspective. A partial and indirect confirmation, also in the sense mentioned above, is provided with the synthetic evaluation of the ranking provided by Forbes 2000 [59] (there is a significant number of companies performing well and registering a somewhat more modest level of annual innovative capacity, even if they do not enter the BCG ranking, namely companies such as Hon-Hai in Taiwan, Caisa bank in Spain, Horton in Mexico, etc.).

Next, in order to identify the extent to which there are certain sub-clusters (small clusters that have similar characteristics based on 2–3 variables considered) within the entire sample of 50 companies, we will proceed to plot a dendogram (Figure 9) and a scree plot to determine the number of clusters (Figure 10). Theoretically, there are two extreme situations (all 50 firms are differentiated and no sub-clusters can be established; all 50 firms are similar and form a single cluster).

**Dendograma**

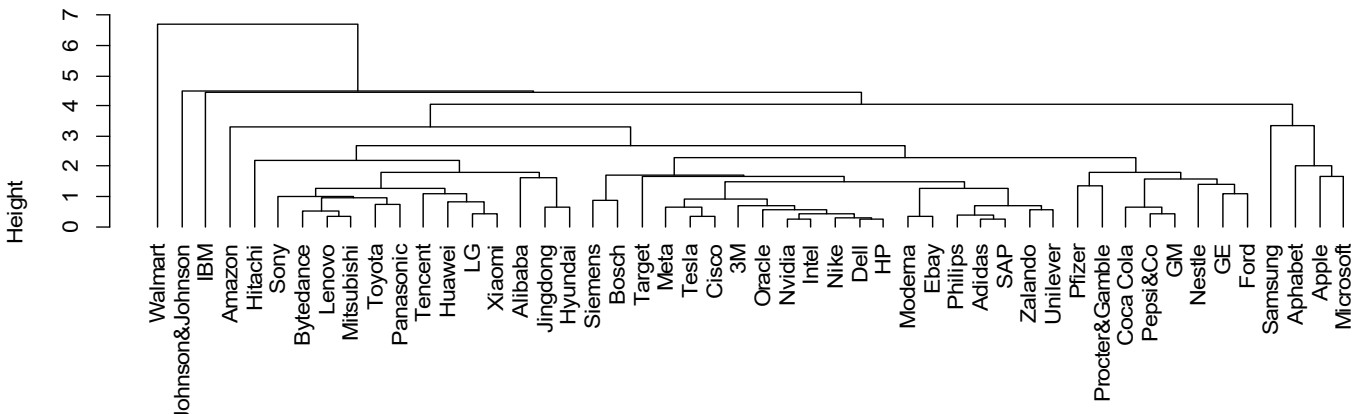

**Figure 9.** Dendogram (classification tree).

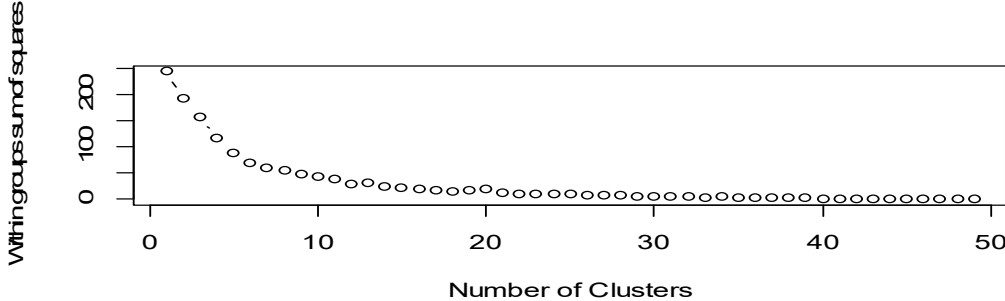

**Figure 10.** Scree plot for determining the optimal number of clusters.

Based on the data in Figures 9 and 10, it follows, depending on the variables considered, that the optimal number of clusters would be between three and five different sub-cluster firms.

In order to highlight more clearly the number of subgroups (distinct clusters that can be constituted for the entire BCG sample), we resorted to the presentation of cluster mapping (Figure 11) and a cluster plot (Figure 12).

From Figures 11 and 12, it is easy to deduce that 3–4 sub-clusters can be formed, depending on the variables by which the firms are compared to each other; each sub-cluster/cluster would, however, have a different number of firms that it is made up of (from about 5 firms to 20 firms per cluster).

The data presented by us previously (Sections 3–6 of the study), together with those shown at this point of the study, lead us to some preliminary conclusions.

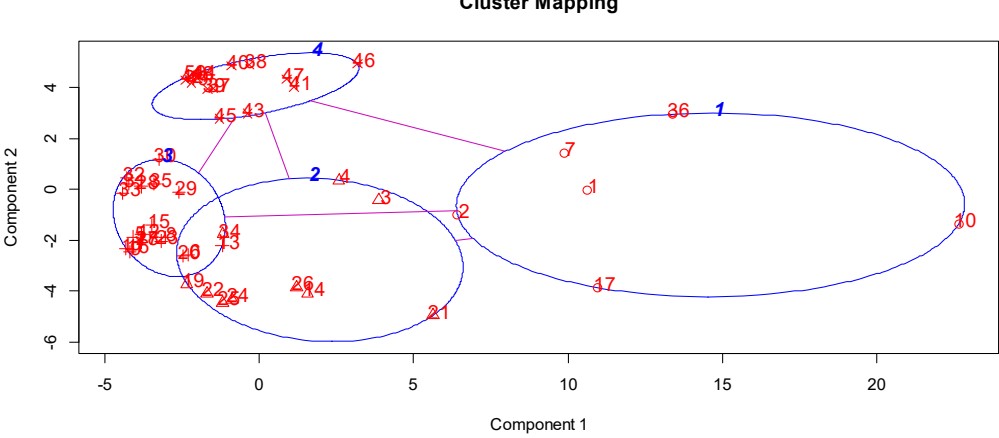

**Figure 11.** Cluster mapping presentation.

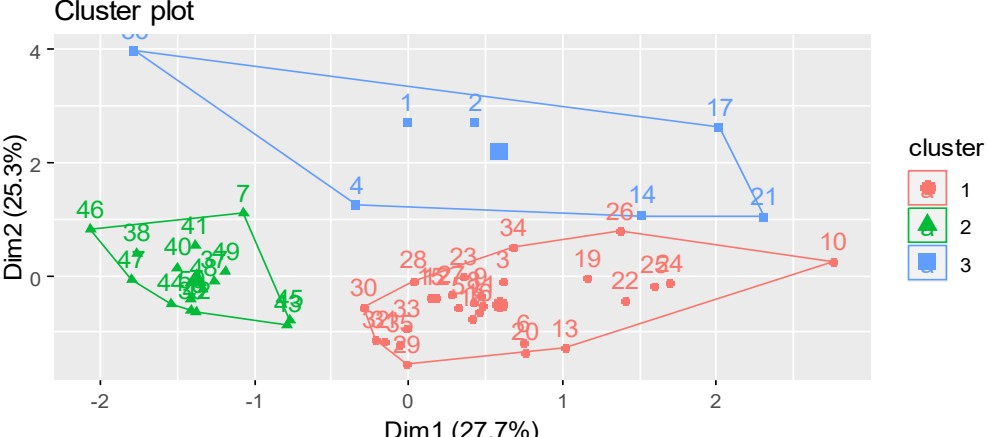

**Figure 12.** Cluster plot presentation.

Following the cross-assessment of financial/social innovation capacity from firms to countries (only as an economic assessment and not as a statistical one), most of the hypotheses, H1–H4, formulated at the beginning of the study are at least partially confirmed:

✓ Hypotheses H1 and H2 formulated by us at the beginning of the study (Section 3.1) are fully confirmed, in the sense that there is no direct association/conditioning/influence relationship between the financial/social innovative capacity and the technical innovative capacity of a firm. As a confirmation of H2, it follows that, whether or not disruptive technologies are used extensively, the financial innovations obtained annually can be empirically evaluated as representing up to about 80% of the social innovations that are achieved by such entities. On the other hand, with regard to hypothesis H3 formulated by us, it appears that this is only partially confirmed, based on "cross-checking" between the BCG studies and the GII and GCI reports, as there are hundreds of thousands of other innovative firms that are not included in any international innovation ranking. However, at the same time, hypothesis H3 is partially confirmed because the world's leading countries that dominate the GCI and GII rankings have firms in the BCG survey, the Forbes survey, and other international rankings at the same time. The H4 hypothesis is also fully confirmed in the sense that disruptive technologies, especially those with financial implications in the range of digital technologies, have become essential not only for fintech firms but for any other company and/or country.

✓ The existing innovative capacity of the world's leading countries (both financial and technical innovation) is reflected in the existence of a number of firms that are large enough to enter various international innovation rankings. Even when looking at the 2000 firms in the Forbes ranking [59], it appears that this ranking is far dominated by American MNCs, then Asian MNCs, and then European MNCs. Simply put, the number of European firms that are sufficiently competitive and innovative has decreased significantly in terms of representation in the BCG survey and/or representation in other international rankings.

✓ There is a fairly strong conditioning from countries to firms on the innovative capacity achieved annually, as countries and various international entities (EU and OECD) directly influence R&D activity through regulations and funds allocated to firms. This means that, theoretically, we will find medium and large firms coming from countries at the middle of the GCI and GII rankings that are not included in the BCG study, but have significant annual financial or technical innovation achievements.

✓ Conversely, the existence of technical and social innovation capacity at the firm level does not automatically/implicitly reflect on innovation capacity at the country level.

✓ Leading US-based companies in various international rankings [59] have a real monopoly on international technical and social innovation. Companies that make it into the BCG rankings and come from European countries are finding it increasingly difficult to maintain their innovative position internationally (Europe is far behind the US and even Japan and China more recently).

✓ With the use of digital/disruptive technologies, any company in industries, commerce, or tourism can be considered innovative at the international level even if it does not obtain its own annual patents (such as Walmart, Zalando, Jingdong, most of fintech, firms, etc.); it can still be extremely innovative in its relationship with the market, its customers, its organizational structures, etc. It is not by chance that tourism companies such as Marriott and Hilton have been found in various BCG rankings as being among the most innovative in terms of funds allocated to digital technologies and networking with various stakeholders (Marriott was also, in 2019, in a similar ranking by Forbes).

✓ The majority of marketing and organizational innovations made annually by firms (whether or not they belong to countries considered to be innovative) will take the form of social innovations of which about up to 80% will be found as financial innovations.

✓ Internationally, there has been a trend in recent decades to set up "international/global-born companies", which means that investors from one or more countries raise capital

and set up successful start-ups, fintech or not, to operate from scratch in various foreign markets [73,74]. In the same vein, we recall "fintech" investments/firms [75], whereby the owners aim to create from scratch a successful start-up in high-tech sectors of the global economy. Such internationally newly created firms, through the strategies applied and the results obtained, seem to deviate from the criteria for gaining competitive advantage that were outlined by Porter [60].

Therefore, it is necessary to conclude that digital technologies are currently generating new paradoxes in the theory of competitive advantage and classical organizational theory. From the perspective of our study, this means that it is not possible to assess globally and/or with large sectors of the economy the number and importance of financial innovations made by firms with disruptive technologies. It is only possible for such an assessment to be made by managers in firms that have KM and social innovation strategies.

*6.3. Principles for Financial Innovation*

Following the issues/arguments we have raised in the first sections of the research (Sections 2–5), as well as the clear and argued distinction between financial innovation and social innovation, a number of theoretical principles can be formulated that could be considered by firms (MNCs and SMEs) to base their strategies on innovation, KM, HR, the use of financial technologies, etc.

First of all, we stress that to improve financial innovation in companies with the help of digital technologies, there must be 2–3 key values at the core of the organizational culture and a single vision at the top management level (in relation to product, technologies, market, and organization). It should be concluded in advance that the realization of technical innovations by firms that are protected with patenting (as shown in Table S2) is not directly conditioned by the realization of financial innovations as part of social innovations (which may or may not be protected using registration with a state agency). At the same time, it appears that the competitive position of a company operating in the manufacturing industry is mainly determined with the stock of the tacit knowledge of employees, such as know-how (technical innovations that can be protected). The financial innovations that can be protected by companies (according to Table S2) are empirically estimated, according to the Pareto principle, to represent about up to 80% of the total social innovations made annually by organizations. Therefore, the largest share of social innovations achieved annually by a firm, i.e., 80% of the total, will be found as financial innovations (what can or cannot be protected using registration within a state agency).

To a large extent (but not totally), both categories of financial innovations refer to innovations related to the market and the organization of a firm. The whole "pyramid construction" that explains financial innovation at the firm level is based on existing values, culture, and education at the country/firm level, as shown in Figure 13. The basic idea in the following figure is that the essential foundation for any kind of innovation (technical vs. social; financial vs. other social innovations; and disruptive vs. business-as-usual innovations) is based on a clear and coherent vision at the top management level of an organization and on the tacit–explicit knowledge stock held by the employees at the bottom of the pyramid and employees who are willing to continuously learn and use creativity to bring about novelties that will then be confirmed by the market. This means a proper KM and HR strategy to motivate, reward, train, and qualify all employees. There are obviously some influences from the macro-social to the organizational level on the development of innovative capabilities (social and technical) at the firm level. However, it is not possible to formulate a clear and reasoned answer from the title of the study: "How do you evaluate financial innovation?". The realities of the acceptance of the fintech concept over the last decade show that, particularly in Asian countries, fintech companies and fintech-based operations have expanded the most rapidly [37]. The vast majority of MNCs in the BCG 2023 study, as well as large banks or insurance companies in the UNCTAD classification [51], are rapidly expanding their financial technology base to adapt to the fintech revolution [14]. One of the preliminary conclusions of our study is that only managers within a company can

assess and/or intuit the more accurate share of financial innovations based on disruptive technologies in the overall social innovation set. This idea is explicitly illustrated by us in the following figure. Within the same graphical scheme in Figure 13, we explicitly highlight the share that fintech-industry-specific financial innovations could reach in the total annual social innovations made by a company.

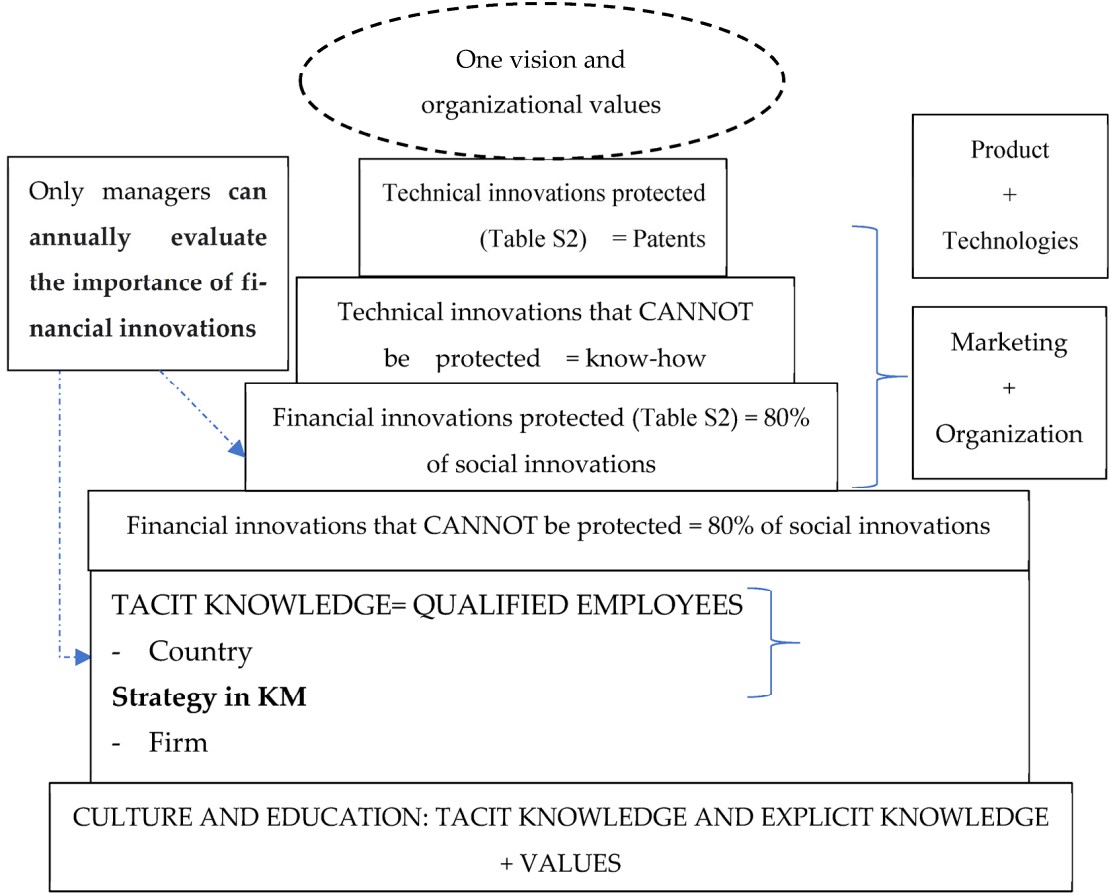

**Figure 13.** Financial innovation as part of social innovation within firms. Source: Authors' design.

In the first part of the study (Section 2. Literature Review), we argued that it is useful to apply the Pareto principle on the random distribution of outcomes generated with an economic variable to the success and/or commercial application of social innovations vs. technical innovations. The same Pareto principle is useful to make a clear distinction between financial and other types of social innovations (up to 80% of social innovations achieved annually by a firm can be included in the category of financial innovations). It is only by measuring the commercial exploitation of a financial innovation based on disruptive technologies that one can more accurately assess/quantify the success of such an innovation. The statement in Figure 13 "Only managers can annually evaluate the importance of financial innovations" is directly related to the basic idea of our study. Simply put, it is not possible to formulate a precise and reasoned answer to the question "How do we evaluate the financial innovations made by firms?", since we are talking about millions of firms in the real economy.

As mentioned earlier, the vast majority of MNCs in the BCG study for 2023 and 2022 are making successful use of various disruptive technologies (digital ones as appropriate, but also some technologies such as non-digital EV batteries and the aforementioned alliance between Tesla, Rivian, Ford, and GM) [76,77]. The disruptive strategy applied by Toyota and then Tesla with respect to exploiting patents to improve EVs clearly has major financial implications, both for companies that are essentially changing practices previously applied

to patents over more than two centuries, and for the thousands of firms that have already taken over various patents from the two companies. Out of the total of around 23,000 EV patents, Toyota has already assigned more than 8000 patents to thousands of companies, including fintech start-ups, to date. Since 2000 or even earlier, companies such as Toyota, IBM, Samsung, Apple, etc., have resorted to organizational innovations with major financial implications for their results/performance over the last two decades. A well-known example is Samsung, which has resorted to major changes in the organizational structure and marketing strategy to overcome the effects of the 1997 Asian crisis [78].

As a result of Samsung's organizational and marketing strategy innovations, it doubled its brand value within 5 years of initiating the reforms [78]. A more precise evaluation of organizational, marketing, and similar innovations, including for the adaptation of large MNCs to the fintech revolution [14,35], can only be performed by the top management of these companies and on medium time intervals (not necessarily annually). The assessment carried out for the 50 companies in the BCG study based on AR 2019–2021 (Table S2 in Supplementary) leads us to the conclusion that the vast majority of these firms have applied and are applying digital technologies massively to improve their financial/social innovation capability. The very concept of social innovation [29] has become a real challenge for large non-financial MNCs such as those in the BCG study. The same is true for financial companies that are included in the UNCTAD rankings, including banks, insurance companies, investment funds, etc. [51]. According to [29], large banks such as Deutsch Bank or City Group have set up units to manage "micro-funds" to support small start-ups and/or people with more modest incomes. The examples cited show that even large financial companies that are not included in the BCG study (analyzed by us) have applied strategies/practices specific to fintech markets since 2000.

The major problem currently faced by large MNCs is to adapt quickly to the new trends imposed by fintech firms and markets [31]. The Forbes 2000 study [45] includes banks, insurance companies, and investment funds; the ranking remains dominated by American firms, but also includes banks/firms from other countries and regions of the world (Canara Bank—Sweden; Axis Bank—India; DNB Bank—Norway; E-L Financial—Canada, etc.). Such banking and credit institutions have started to make massive use of financial technologies in order to adapt to the social innovation imposed by modern society and to resist the competition with fintech firms [30]. The concept of social innovation is, argues [29]. "under-theorized" even though there are hundreds of papers/studies on it; it has become a real challenge for Europe over the last decade as the fintech revolution manifests itself. Our proposed study sheds some light on the relationship between financial and other social innovations amidst the use of disruptive technologies, as well as on the relationship between the organizational and societal perspective of social/financial innovation. In addition to what has been shown in this section of the study, we list below 10 principles that are "de facto" at the core of financial innovation with disruptive technologies over the last two decades at the global level.

✓ Principle 1 (P1): Using disruptive technologies

The first principle is that any business (MNCs and SMEs), as well as other types of organizations, can and should make extensive use of disruptive/digital technologies to improve their financial innovation capacity as part of their social innovations. One of the directions of innovation strategy is towards open innovation by setting up various business networks for innovation.

✓ Principle 2 (P2): Applying strategies in KM

Every company should have a clear and distinct strategy for KM and continuous innovation. The focus of this strategy should be on technical and/or social innovation, depending on the field/sector in which a company operates. The best theoretical approaches can be taken towards social innovation in relation to different stakeholders.

✓ Principle 3 (P3): Stakeholder orientation

Any company has, by the very nature of its daily activity, a certain volume of financial relations with a group of stakeholders (suppliers, customers, employees, shareholders, public institutions, etc.). These day-to-day financial relationships are nowadays carried out through various solutions offered by ITC and other disruptive technologies. The strengthening of financial stakeholder relationships needs to be continuously monitored and improved as new tools and disruptive technologies emerge, especially from the artificial intelligence (AI) category.

✓ Principle 4 (P4): Multiplying financial innovations

When a company/organization has a KM strategy and systematically uses various digital/disruptive technologies, it can exploit the multiplier effect of an innovation. This means that a financial innovation based on digital technologies achieved in one direction of action (e.g., in relation to suppliers) can then be extended/duplicated very quickly in another direction of action/interest of a firm (e.g., in relation to customers/consumers). According to Drucker's arguments, customers and the market should be the most important source of inspiration for a firm in its attempt to achieve financial innovation by acquiring new knowledge and managing this knowledge through the use of various technologies [64].

✓ Principle 5 (P5): Customer/market orientation

The most important element/factor that should be at the heart of a continuous innovation strategy in the financial sector using various technologies remains the customer and the market, including other competitors. In other words, the customer and the market have become, over the last three decades, the "core" element of any strategy applied by firms in KM and continuous innovation. The continuous innovation of the "financial innovation" type remains the most important part of the social innovation undertaken/achieved by firms; the former cannot be "broken/dissociated" from the broader framework of social innovation [21]. Whatever the size of a firm and whatever the area of location, the starting/foundation point of KM strategy and continuous innovation is given with the values that customers/consumers believe in. Only by starting from this point and building ICT-based networks for open innovation can firms strengthen their competitive position and innovative capacity in a given market. It is only the satisfied customer that determines the outcome and/or existence of a firm [64].

✓ Principle 6 (P6): Financial innovation is partially associated with the business area

Most financial innovation will be related to process, marketing, and organizational innovation. However, the chances of achieving such innovations are conditioned by the location of firms in large sectors (from high-tech to low-tech).

✓ Principle 7 (P7): Disruptive technologies diminish the importance of size

The widespread use of digital and other technologies tends to eliminate/diminish the importance of the size of a firm in all cases where the KM strategy focuses on financial innovation as part of social innovation. Only a fraction of financial/social innovations will prove, over time, to be disruptive in the market and society.

✓ Principle 8 (P8): Financial innovation is decoupled from technical innovation

The existing vision at the company level must clearly differentiate between financial/social innovation and technical innovation; there are companies that have a large number of patents obtained annually, but have an insignificant position on social innovation that can be protected (as shown in Figure 11).

✓ Principle 9 (P9): Innovation in firms should be systematic

Firms aiming to become innovative need to build their continuous/systematic innovation strategies (according to the existing EU and countries rules, as well as the most innovative firms' practices) with regard to KM, MRU, and proposed targets as a result of innovation.

✓ Principle 10 (P10): Some financial innovations will be disruptive

Provided that a firm has a systematic innovation strategy, it follows that we can talk about sustaining and disruptive innovations [28]; not all the novel elements brought by a firm in its financial relationships with various stakeholders will turn out to be immediately disruptive. This implicitly means orienting the firm towards open innovation networks for which management optimization requires various ITC solutions [24].

## 7. Conclusions

The comparative analysis based on Global Innovation Index and Global Competitiveness Index data vs. the BCG ranking shows significant changes in the strategies applied by countries/firms regarding technical and social innovations/inventions carried out by different entities (researchers, firms, universities, etc.). However, it is difficult to assess the situation of social/financial innovations made by the same countries and firms in the two contexts of a global crisis or recession. This is even if, as we highlighted above, the GII and GCI components also include indicators such as business models developed by firms, the relationship with the market, etc. Our assessments based on BCG 2022–2023 lead us to conclude that there are certain common elements between the financial innovative capacity in firms and the same innovative capacity at the country level, and that at both levels of analysis, disruptive technologies have become the essential supporting/enhancing factors.

The question formulated in the title of the study "How do we evaluate financial innovations?" is predominantly rhetorical, as the realities encountered in various financial markets, whether associated with fintech markets or not, are extremely diverse. In the same sense, the existing conceptualizations in the literature on the use of financial technologies at the firm and/or country level are made from significantly different perspectives (banks, traditional industries, SMEs, sociology, digital technologies, etc.). The major changes brought about by fintech firms and markets over the last decade at the global level will, we believe, generate significant reconceptualization that will occur in organizational theory, start-up financing, innovation strategies, etc. One of the conclusions of our study is that digital and other disruptive technologies offer major opportunities for researchers, as well as MNCs' and SMEs' firms and managers, to build effective KM and social innovation strategies. Another conclusion of the study is that the fintech revolution in the use of digital technologies to create new markets is starting to decisively influence the innovation strategies of established MNCs in almost all sectors of the global economy (industry, distribution, telecommunications, banking, insurance, etc.). It is difficult to predict precisely in which directives and how future trends will manifest themselves in the strategies applied by thousands of fintech firms and/or traditional firms involved in fintech operations. The present study, however, makes some clear arguments that the fintech revolution is only at its beginning, that it requires new regulations/rules adopted by governments/international institutions, and that it will generate social benefits for millions of people. In order to overcome the specific economic challenges of the 2020–2021 period, social innovations have become more necessary than ever for any type of business organization. Among other conclusions our study leads to, we mention the 10 principles that companies can consider to build their KM and continuous social innovation strategies.

The post-2000 literature on the concept of "disruptive innovation", "open innovation", "financial innovation", "social innovation", etc., undoubtedly creates some confusion when companies propose strategies for strengthening/orienting their culture towards continuous innovation. It is necessary and useful for firms, we believe, to distinguish as clearly as possible, both conceptually/theoretically and pragmatically, between technical innovations and social innovations. Of course, technical innovations (also called sustaining and/or disruptive innovations) will remain a major, essential objective for companies in various industrial sectors. On the other hand, firms operating in various service sectors (banking, insurance, retailing, transport, consultancy, health, etc.) can, at best, make more intensive use of existing technologies to strengthen their market position, which requires precisely that social/financial innovations be made through the use of equipment/technology. As we have shown, there are theoretically three 'driving forces' (technologies, managers, and

competition) that can jointly support the improvement in innovative capabilities at the firm/country level. The present study is only a preliminary one on the complex relationship between financial/social innovation capacity within firms and the same innovation capacity at the country level. It is theoretically useful as it provides a starting point for other similar studies that aim to present a clear/consistent picture of the role of disruptive technologies in improving innovative capacity in firms and countries alike. The study is also useful from a pragmatic perspective, as it proposes/suggests small points of support for managers in any type of firm and any country in their attempt to build effective strategies on financial/social innovation. In future studies, the authors aim to expand the sample of firms that are internationally significant (considering 200–300 firms that are nominated in various rankings) and on the basis of a thorough assessment of technical and social innovative capacity to draw further conclusions that explain innovative capacity at the country level and would support policy makers in formulating macro-economic strategies on education, R&D, research infrastructure, etc.

**Supplementary Materials:** The following supporting information can be downloaded at: https://www.mdpi.com/article/10.3390/fintech2030033/s1. Table S1. Preliminary Statistical Data for GCI, Selective for Seven Sub-pillars (Total GCI Score and Seven Existing Sub-pillars in 2019); Table S2. Summary of BCG Companies According to Social/Financial and Technical Innovation, Period of 2004–2022 (Assessment at the Time of 2022 Based on Annual Report).

**Author Contributions:** Each author made contributions to the conception and design of the work, and the acquisition, analysis, and interpretation of data. All authors have read and agreed to the published version of the manuscript.

**Funding:** This research received no external funding.

**Institutional Review Board Statement:** The study was conducted in accordance with the Declaration of Helsinki.

**Informed Consent Statement:** Not applicable.

**Data Availability Statement:** All data can be shared with other researchers; data used can be found in international rankings and the final list of references.

**Acknowledgments:** We thank the editor and reviewers for their support in improving the quality of this study.

**Conflicts of Interest:** The authors declare no conflict of interest.

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
