# Peer review of "The Study of the Relationship among GCI, GII, Disruptive Technology, and Social Innovations in MNCs: How Do We Evaluate Financial Innovations Made by Firms? A Preliminary Inquiry"

_fintech, doi:10.3390/fintech2030033_

Round 1
Reviewer 1 Report
Please see PDF file attached.

Needs proofreading and improvements.
Author Response
Attached you have our comments; the new version of manuscript has been upload to the editor.

Reviewer 2 Report
This is a dynamic field of research. Consider updating your literature review to the last 5 years.
Author Response
Attached you have our answer; the revised manuscript has been submited to the editor.

Reviewer 3 Report
This paper investigates the role of disruptive/digital technologies in financial innovation strategies as part of social innovations at both firm and country level. Through analyzing Global Competitiveness Index, Global Innovation Index and 50 innovative multinational companies in the world, the authors argue that they synthesized the factors explaining technical/social innovative capacity and found the driving forces behind financial innovativeness at firm level. Overall, the research focus is not clear; empirical analyses are not conducted properly. Let me explain in detail:
In the introduction section, as a part of the motivation for the study, the authors argue that “in a chaotic business environment, the use of digital technologies as part of disruptive technologies can assistance firms to improve their technical and social innovative capacity, and thus better respond to the challenges of going through a downswing in the cyclical evolution of business.” But the authors did not state how digital technologies improves firms’ technical and social innovative capacity. The key influential channel is not explained.
In the hypothesis design section, H1 does not add much to our understanding of the interactions between technical and social innovative capacity, because failing to find the association in the current data does not mean non-existence of the association in general. H2 lacks support from extant literature, it is merely a retrace of data distributions in the empirical analysis. H3 is not well stated. At country level, there are a lot of confounding factors, some of which are not observable to the authors. We cannot attribute the resources allocation to R&D and technical, social and financial innovation to country level competitiveness.
The methodology of the paper is flawed. Statistical correlation does not suggest causality (e.g., “driving force”) as claimed by the authors. Endogeneity problems are not resolved or even discussed. Summary statistics are missing. The basic data characteristics are not described. Robustness checks on the empirical analyses are missing.
Economic significance of analytical results and policy implications are not well stated.
English is fine in general.
Author Response
Attached you have our answer; the revised manuscript has been submitted to the editor.
